# A Theory of How Pretraining Shapes Inductive Bias in Fine-Tuning

**Nicolás Anguita** [1]
**Francesco Locatello** [2]   **Andrew M. Saxe** [3]   **Marco Mondelli** [2]   **Flavia Mancini** [1]
**Samuel Lippl** [* 4]   **Clémentine Dominé** [* 2]

## Abstract

Pretraining and fine-tuning are central stages in modern machine learning systems. In practice, feature learning plays an important role across both stages: deep neural networks learn a broad range of useful features during pretraining and further refine those features during fine-tuning. However, an end-to-end theoretical understanding of how choices of initialization impact the ability to reuse and refine features during fine-tuning has remained elusive. Here we develop an analytical theory of the pretraining–fine-tuning pipeline in diagonal linear networks, deriving exact expressions for the generalization error as a function of initialization parameters and task statistics. We find that different initialization choices place the network into four distinct fine-tuning regimes that are distinguished by their ability to support feature learning and reuse—and therefore by the task statistics for which they are beneficial. In particular, a smaller initialization scale in earlier layers enables the network to both reuse and refine its features, leading to superior generalization on fine-tuning tasks that rely on a subset of pretraining features. We demonstrate empirically that the same initialization parameters impact generalization in ResNets trained on CIFAR-100 and SVHN as well as Transformers trained on modular arithmetic tasks. Overall, our results demonstrate analytically how data and network initialization interact to shape fine-tuning generalization, highlighting an important role for the relative scale of initialization across different layers in enabling continued feature learning during fine-tuning.

* Co-senior authors. [1]Department of Engineering, University of Cambridge [2]Institute of Science and Technology, Austria (ISTA) [3]Gatsby Computational Neuroscience Unit and Sainsbury Wellcome Centre, UCL [4]Center for Theoretical Neuroscience, Columbia University. Correspondence to: Nicolás Anguita <na658@cam.ac.uk>.

*Proceedings of the 43rd International Conference on Machine Learning*, Seoul, South Korea. PMLR 306, 2026. Copyright 2026 by the author(s).

## 1. Introduction

Deep neural networks are often trained in stages: first, they are pretrained on general-purpose tasks for which large amounts of data are available; then, they are fine-tuned on the actual target task, for which data may be more limited. This pretraining–fine-tuning pipeline (PT+FT) has emerged as an essential workhorse for modern deep learning (Bommasani, 2021; Parthasarathy et al., 2024; Awais et al., 2025).

To benefit from fine-tuning, neural networks must learn task-specific features during pretraining and reuse them during fine-tuning (Lampinen & Ganguli, 2019; Saxe et al., 2019; Shachaf et al., 2021; Tahir et al., 2025). At the same time, fine-tuning further adapts these features, potentially improving generalization (Yosinski et al., 2014; Huh et al., 2016; Jain et al., 2024). As a result, the success of the PT+FT pipeline critically depends on how feature learning unfolds over the pretraining and fine-tuning stages.

Despite the practical success of PT+FT, the factors governing feature learning across pretraining and fine-tuning have remained unclear. Prior work has shown that weight initialization—in particular, its absolute and relative scale across layers—shifts neural network training between a fixed-feature ("lazy") and a feature-learning ("rich") regime (Chizat et al., 2019; Braun et al., 2022; Dominé et al., 2024; Kunin et al., 2024). In the single-task setting, feature-learning regimes often induce improved generalization (Chizat & Bach, 2020; Fort et al., 2020; Vyas et al., 2023). However, it remains unclear how these insights extend to multi-stage training like PT+FT (though see Lippl & Lindsey, 2024, see Section 2). This limits our ability to use weight initialization as a principled tool for controlling the balance between reusing pretrained features and inducing continued feature learning.

To address this gap, we study the PT+FT pipeline in diagonal linear networks, a tractable theoretical model that exhibits feature-learning behavior, and develop an end-to-end theory of the resulting generalization error. Our theory elucidates how weight initialization affect the inductive bias of PT+FT, and how this bias interacts with data properties to determine when different initialization schemes are optimal.

**Specific Contributions.**

- We analytically derive the implicit bias of PT+FT in diagonal linear networks and the resulting generalization error as a function of weight initialization, task parameters, and data scale (Section 4). To our knowledge, prior work has not provided such an integrated characterization in a PT+FT setup.

- We identify three limiting regimes induced by this implicit bias: (I) a pretraining-independent rich regime, (II) a pretraining-dependent lazy regime, and (III) a pretraining-independent lazy regime. We further highlight a universal trade-off between pretraining dependence and feature learning in fine-tuning (Section 5.1).

- In light of this trade-off, we highlight a fourth, intermediate regime: (IV) the pretraining-dependent rich regime. We find that the relative scale of initialization plays a key role in entering this regime (Section 5.2).

- We demonstrate that pretraining and fine-tuning task parameters determine which of these regimes will generalize best (Section 5.3). In particular, we highlight the relative scale of initialization across layers as a key lever for controlling the inductive bias of PT+FT.

- Finally, in Section 6, we demonstrate that our theoretical insights extend empirically to deep neural networks trained on vision tasks or modular addition.

Altogether, we provide an end-to-end understanding of how initialization choices shape fine-tuning generalization. In doing so, we identify actionable levers for controlling the interplay between reusing pretrained features, refining those features, and learning new features.

## 2. Related Work

**Rich and Lazy Learning.** Neural networks are often trained in an overparameterized setting where there are many possible solutions to the training data. In this setting, the training procedure consistently biases the network towards a particular solution (a phenomenon called "implicit regularization," Soudry et al., 2018), explaining why neural networks generalize well even without explicit regularization (Zhang et al., 2017). The distinction between lazy and rich learning regimes has become central to understanding the implicit regularization of neural networks. In the lazy regime, training dynamics are governed by a fixed representation, causing the network to effectively behave like a kernel machine (Jacot et al., 2018; Chizat et al., 2019). In contrast, the rich (or feature-learning) regime is marked by the emergence of task-specific, low-dimensional representations and is commonly linked to finite-width networks and

smaller weight scales (Saxe et al., 2013; Gunasekar et al., 2018; Savarese et al., 2019; Lyu & Li, 2020; Nacson et al., 2019; Chizat & Bach, 2020; Atanasov et al., 2021). Recent work has shown that the relative and absolute scale of initial weights across layers induce a continuum of feature-learning dynamics and corresponding inductive biases (Woodworth et al., 2020; Azulay et al., 2021; Dominé et al., 2024; Kunin et al., 2024). Our work builds on these insights to show how this spectrum manifests in the PT+FT setting. We show that, depending on task parameters, intermediate regimes can induce superior generalization, suggesting that analyses restricted to extreme cases may miss important behaviors.

**Implicit Regularization and Diagonal Linear Networks.** Diagonal linear networks (Section 3.1) instantiate a simple, analytically tractable model that captures the changes in inductive bias arising from different learning regimes in more complex models: specifically, lazy-regime training finds the (non-sparse) solution with minimal $\ell_2$-norm, whereas rich-regime training finds the sparse solution with minimal $\ell_1$-norm (Woodworth et al., 2020; Azulay et al., 2021). Prior work has characterized the emergence of the infinite-time solution in these networks (Pesme & Flammarion, 2023; Berthier, 2023) and investigated the role of stochasticity and step size (Pesme et al., 2021; Nacson et al., 2022).

**Implicit Regularization of Fine-Tuning.** In dense linear networks, a higher similarity in task features between pretraining and fine-tuning improves generalization performance (Lampinen & Ganguli, 2019; Shachaf et al., 2021; Tahir et al., 2025). Moreover, fine-tuning with backpropagation can result in improved in-distribution generalization compared to only training the linear readout, but may harm out-of-distribution performance (Kumar et al., 2022; Tomihari & Sato, 2024). These theories generally focus on single-output dense linear networks (which have the same inductive bias in the rich and lazy regime) or training in the kernel regime (Malladi et al., 2023). In contrast, our perspective allows us to characterize how different pretraining initializations shift the fine-tuning behavior between the lazy and rich regime. Other works characterize the influence of pretraining data selection on fine-tuning under distributional shifts (Kang et al., 2024; Cohen-Wang et al., 2025; Jain et al., 2025). Finally, Lippl & Lindsey (2024) characterized the inductive bias of diagonal linear networks for infinitesimal initial weights in pretraining and studied the resulting generalization behavior using simulations in a teacher-student setup. Here we substantially extend their work by characterizing the inductive bias conferred by a broad range of pretraining and fine-tuning initializations, revealing a crucial role played by the relative scale of initialization. Moreover, we derive an analytical theory describing the generalization error (rather than relying on simulations), providing closer insight into specific conditions under which certain pretraining initializations help or hurt performance.

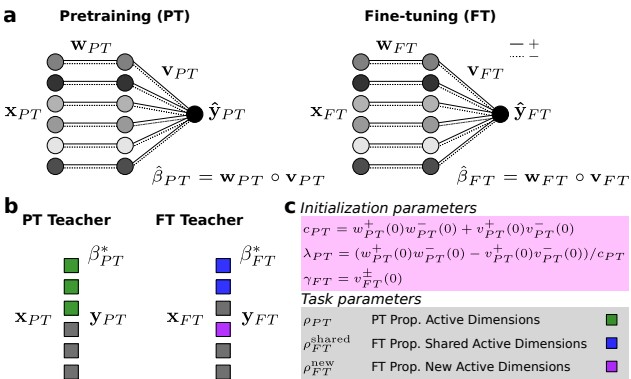

*Figure 1.* **Setup.** (a) We pretrain and fine-tune diagonal linear networks. (b) We generate pretraining and fine-tuning tasks by sampling from teacher networks with a subset of active dimensions, which can overlap between tasks. (c) Our theory considers the influence of initialization parameters $c_{PT}$, $\lambda_{PT}$, and $\gamma_{FT}$ and task parameters $\rho_{PT}$, $\rho_{FT}^{\text{shared}}$, and $\rho_{FT}^{\text{new}}$.

**Replica-Theoretical Characterization of the Generalization Error.** We leverage the replica method (Edwards & Anderson, 1975; Mézard et al., 1987) to characterize the ability of diagonal linear networks to recover ground-truth parameters in a fine-tuning setup. The replica method is non-rigorous, but provides a powerful tool for analytically characterizing estimation errors in the high-dimensional limit under different penalties, including $\ell_1$ (Guo & Verdú, 2005; Rangan et al., 2009). These formulas were later confirmed using rigorous approaches (Bayati & Montanari, 2011a;b). We specifically rely on the approach in Bereyhi & Müller (2018); Bereyhi et al. (2019), which characterizes estimation errors with potentially different penalties across dimensions. Our results imply that pretraining can be understood as modifying the penalties across different dimensions (see also Khajehnejad et al., 2009; Vaswani & Lu, 2010).

## 3. Theoretical Setup

### 3.1. Network Architecture

We study the generalization behavior of *diagonal linear networks* (**Fig. 1**), a simple one-hidden-layer network architecture that parametrizes linear maps $f : \mathbb{R}^D \to \mathbb{R}$ as

$$f_{\mathbf{w},\mathbf{v}}(\mathbf{x}) = \beta(\mathbf{w},\mathbf{v})^T x, \tag{1}$$

where $\beta(\mathbf{w},\mathbf{v}) := \mathbf{v}^+ \circ \mathbf{w}^+ - \mathbf{v}^- \circ \mathbf{w}^- \in \mathbb{R}^D$ ($\circ$ indicates element-wise multiplication). Here, $\mathbf{w}^+, \mathbf{w}^- \in \mathbb{R}^D$ comprise the hidden weights of the network and $\mathbf{v}^+, \mathbf{v}^- \in \mathbb{R}^D$ comprise its output weights. $\mathbf{w}^+, \mathbf{v}^+$ and $\mathbf{w}^-, \mathbf{v}^-$ parameterize the positive and negative pathway, respectively; separating these pathways allows us to initialize the network at $\beta = 0$ while avoiding a saddle point.

Diagonal linear networks capture two key aspects of the deep neural networks used in practice: 1) their initial

weights control their transition between the rich (feature-learning) and lazy (kernel) regime and 2) their generalization behavior is different between these regimes. They therefore provide a useful theoretical model to study the impact of initialization on pretraining and fine-tuning.

Beyond these high-level motivations, diagonal linear networks are also mathematically connected to nonlinear architectures. The learning dynamics of ReLU networks decompose into two orthogonal components: one capturing changes in the normalized first-layer weight vector, and one capturing changes in the magnitudes of the first and second layer, with the latter being identical to the dynamics of a diagonal linear network. In particular, accordingly, both diagonal linear networks and ReLU networks exhibit a sparse inductive bias in the rich regime (Woodworth et al., 2020; Chizat & Bach, 2020). This grounds the relevance of our analysis for nonlinear settings.

### 3.2. Network Training

We investigate diagonal linear networks trained with gradient flow to minimize mean-squared error. We consider two datasets: a pretraining (PT) task $\mathbf{X}_{PT} \in \mathbb{R}^{N_{PT} \times D}, \mathbf{y}_{PT} \in \mathbb{R}^{N_{PT}}$ and a fine-tuning (FT) task $\mathbf{X}_{FT} \in \mathbb{R}^{N_{FT} \times D}, \mathbf{y}_{FT} \in \mathbb{R}^{N_{FT}}$. The training proceeds in two stages: we first pretrain on the PT task and then fine-tune on the FT task (PT+FT) (**Fig. 1a**). In both cases, we assume that the model trains to zero training error. We denote the parameters learned during pretraining and fine-tuning by $\mathbf{w}_{PT}(t), \mathbf{v}_{PT}(t)$ and $\mathbf{w}_{FT}(t), \mathbf{v}_{FT}(t)$, the corresponding network functions by $\hat{\beta}_{PT}(t)$ and $\hat{\beta}_{FT}(t)$, and their predictions by $\hat{\mathbf{y}}_{PT} = \mathbf{X}_{PT}\hat{\beta}_{PT}$ and $\hat{\mathbf{y}}_{FT} = \mathbf{X}_{FT}\hat{\beta}_{FT}$. $t$ denotes the learning time, $t = 0$ indicating the initial weights, and $t = \infty$ the weights at the end of training. Where the context is clear, we denote the network functions at the end of pretraining and fine-tuning by $\hat{\beta}_{PT}$ and $\hat{\beta}_{FT}$.

**Initialization.** We assume $w^+(0) = w^-(0)$ and $v^+(0) = v^-(0)$ (to avoid biasing the network towards positive or negative coefficients). Building on insights from the feature learning literature, we focus on two key aspects of the weight initialization: 1) the **absolute scale** (capturing the overall initial weight magnitude),

$$c_{PT} := \mathbf{w}_{PT}^+(0)\mathbf{w}_{PT}^-(0) + \mathbf{v}_{PT}^+(0)\mathbf{v}_{PT}^-(0), \tag{2}$$

and 2) the **relative scale** (capturing the difference in initial weight magnitude between the first and the second layer),

$$\lambda_{PT} := \frac{\mathbf{w}_{PT}^+(0)\mathbf{w}_{PT}^-(0) - \mathbf{v}_{PT}^+(0)\mathbf{v}_{PT}^-(0)}{c_{PT}} \in [-1, 1]. \tag{3}$$

After pretraining, we re-balance the positive and negative pathways by setting

$$\mathbf{w}_{FT}^\pm(0) := \mathbf{w}_{PT}^+(\infty) + \mathbf{w}_{PT}^-(\infty), \tag{4}$$

to ensure that the effective network function at the beginning of fine-tuning is zero again: $\beta_{FT}(0) = 0$.[1]

We re-initialize the readout weights to a fixed scale $\gamma_{FT}$:

$$\mathbf{v}_{FT}^{\pm}(0) := \gamma_{FT} \geq 0. \qquad (5)$$

By systematically varying these three initialization parameters ($c_{PT}$, $\lambda_{PT}$, and $\gamma_{FT}$), we aim to understand their effect on fine-tuning performance.

### 3.3. Data Generative Model

To investigate the generalization performance of diagonal linear networks trained in this manner, we consider a *teacher-student setup*, where we sample ground-truth pretraining and fine-tuning functions ("teachers") by sampling from a joint distribution $\beta_{PT}^*, \beta_{FT}^* \sim p_{PT,FT}(\beta_{PT}^*, \beta_{FT}^*)$. We generate the dataset by sampling random inputs, $\mathbf{X}_{PT}, \mathbf{X}_{FT} \sim \mathcal{N}(0, \frac{1}{D})$, and generating the outputs through the teachers, $\mathbf{y}_{PT} = \mathbf{X}_{PT}\beta_{PT}$, $\mathbf{y}_{FT} = \mathbf{X}_{FT}\beta_{FT}$. We aim to characterize the deviation between the ground-truth $\beta_{FT}^*$ and the estimate resulting from fine-tuning, $\hat{\beta}_{FT}$:

$$\mathcal{E} := \|\beta_{FT}^* - \hat{\beta}_{FT}\|_2^2. \qquad (6)$$

We characterize $\mathcal{E}$ through a replica-theoretical approach, which characterizes network behavior in the high-dimensional limit ($D \to \infty$), where we scale the pretraining and fine-tuning data size with $D$:

$$\alpha_{PT} := \lim_{D\to\infty} N_{PT}/D, \quad \alpha_{FT} := \lim_{D\to\infty} N_{FT}/D. \qquad (7)$$

For most of our analysis we focus on the case $\alpha_{PT} \geq 1$, in which pretraining perfectly recovers the ground truth ($\hat{\beta}_{PT} = \beta_{PT}^*$). We relax this assumption and analyze imperfect pretraining ($\alpha_{PT} < 1$) in Section 5.3. We call $\alpha_{FT}$ (also known as the "load") the *data scale*. Finally, we characterize a specific family of generative models $p_{PT,FT}(\beta_{PT}^*, \beta_{FT}^*)$, in which there are $J$ underlying groups that are sampled with probabilities $\pi \in \mathbb{R}^J$ for each dimension $d$. For the sampled group $j \in \{1, \dots, J\}$, $\beta_{PT,d}^*$ and $\beta_{FT,d}^*$ are independently sampled from group-specific distributions $p_{PT}^{(j)}$ and $p_{FT}^{(j)}$, i.e. their dependency is mediated via their group membership (see Definition C.1).

To investigate the consequences of our theory (and confirm it in empirical simulations), we will focus on a **spike-and-slab** distribution for $\beta_{PT}^*, \beta_{FT}^*$. In diagonal linear networks, feature learning is useful when only a sparse subset of dimensions are active for a given task. We therefore sample

---

[1]This rebalancing step is motivated by the structure of diagonal linear networks, in which the signs within each pathway are conserved throughout training. Without rebalancing, an additional inductive bias towards features of the same sign as during pretraining would arise, obscuring the effect of feature overlap.

the set of active dimensions for the pretraining task,

$$\theta_{PT} \sim \text{Bernoulli}(\rho_{PT}), \quad \theta_{PT} \in \{0, 1\}^D, \qquad (8)$$

and then generate $\beta_{PT}$ by sampling random signs for the active dimensions (i.e. any $d$ for which $\theta_d = 1$):

$$\beta_{PT} = (\sigma/\sqrt{\rho_{PT}}) \circ \theta_{PT}, \quad \sigma \sim \text{Cat}(\{-1, 1\}). \qquad (9)$$

We similarly assume that a subset of dimensions is active on the fine-tuning task by sampling $\theta_{FT} \in \{0, 1\}^D$. However, in this case, whether a given dimension is active can depend on whether it was already active on the pretraining task:

$$\begin{aligned} \theta_{FT}|\theta_{PT} = 1 &\sim \text{Bernoulli}(\rho_{FT}^{\text{shared}}/\rho_{PT}), \\ \theta_{FT}|\theta_{PT} = 0 &\sim \text{Bernoulli}(\rho_{FT}^{\text{new}}/(1 - \rho_{PT})), \\ \beta_{FT} = b_{FT} \circ \theta_{FT}, \; b_{FT} &\sim \mathcal{N}(0, \tfrac{1}{\rho_{FT}^{\text{shared}}+\rho_{FT}^{\text{new}}}). \end{aligned} \qquad (10)$$

Thus, $\rho_{FT}^{\text{shared}} \geq \rho_{FT}^{\text{new}}$ means that dimensions active during pretraining are more likely to be active during fine-tuning, $\rho_{FT}^{\text{shared}} \leq \rho_{FT}^{\text{new}}$ means that they are less likely (**Fig. 1b**).

Taken together, we will characterize how 1) initialization parameters, $c_{PT}$, $\lambda_{PT}$, and $\gamma_{FT}$, 2) task parameters, $\rho_{PT}$, $\rho_{FT}^{\text{shared}}$, and $\rho_{FT}^{\text{new}}$, and 3) data scale, $\alpha_{FT}$, impact generalization during fine-tuning (**Fig. 1c**).

## 4. Theoretical Characterization of Generalization Error in Fine-Tuning

We will derive a theoretical characterization of the generalization error $\mathcal{E}$ in two steps: first, we will derive the implicit regularization induced by PT+FT in diagonal linear networks as a function of initialization parameters (Theorem 4.1), then we will characterize the resulting generalization error for our data generative model (Proposition 4.2).

### 4.1. Implicit Inductive Bias of Fine-Tuning

We first derive the implicit inductive bias of overparameterized diagonal networks on PT+FT, building on analysis in prior work (Azulay et al., 2021; Lippl & Lindsey, 2024).

**Theorem 4.1** (Implicit bias). *Consider a diagonal linear network trained on PT+FT (as described in Section 3.2). Then, the gradient flow solution at convergence is given by*

$$\arg\min_{\beta_{FT}} Q_k(\beta_{FT}) \quad s.t. \; \mathbf{X}_{FT}^\top \beta_{FT} = \mathbf{y}_{FT}, \qquad (11)$$

$$where \quad Q_k(\beta_{FT}) = \sum_{d=1}^{D} q_{k_d}(\beta_{FT,d}), \qquad (12)$$

$$q_k(z) = \frac{\sqrt{k}}{4}\left(1 - \sqrt{1 + \frac{4z^2}{k}} + \frac{2z}{\sqrt{k}}\operatorname{arcsinh}\left(\frac{2z}{\sqrt{k}}\right)\right), \qquad (13)$$

$$k_d = \left(2c_{PT}(1+\lambda_{PT})\left(1 + \sqrt{1 + (\hat{\beta}_{PT,d}/c_{PT})^2}\right) + \gamma^2\right)^2. \qquad (14)$$

*Proof.* Azulay et al. (2021) proved that $c_{PT}$ (when defined as $c_{PT} = w^+ w^- + v^+ v^-$) and $\lambda_{PT}$ are conserved throughout training (see also Marcotte et al., 2023). Extended calculations allow us to recover the individual layers' weights from the effective network weights after pretraining. Details are relegated to Appendix B.1 due to space constraints. □

Theorem 4.1 shows that the function selected by the network depends on $\lambda_{PT}$, $c_{PT}$, and $\gamma_{FT}$ in a non-trivial manner. Importantly, this implicit bias also depends on $\hat{\beta}_{PT}$, implying that the learned solution potentially depends on the pretraining task. Notably, even though in diagonal linear networks, the relative scale $\lambda_{PT}$ does not impact the inductive bias of pretraining, it does impact the learned hidden representation and therefore the inductive bias of fine-tuning. In Section 5.1 we show how these dependencies shape the resulting learning regime. This substantially extends results by Lippl & Lindsey (2024), which focused on $c_{PT} \to 0$ with $\lambda_{PT} = 0, \gamma_{FT} = 0$. We will show that this only captures a subset of the range of learning regimes we identify. In particular, and perhaps surprisingly, we will see that even for very small $c_{PT}$, different values for $\lambda_{PT}$ can fundamentally change the generalization behavior.

## 4.2. Replica Theory of the Generalization Error

The replica method is a tool from statistical physics that characterizes the typical behavior of high-dimensional estimation problems in the limit $D \to \infty$ (Edwards & Anderson, 1975; Mézard et al., 1987). Rather than analyzing the full joint distribution of estimates $\hat{\beta}_{FT} \in \mathbb{R}^D$, the method reduces the problem to a scalar denoising problem governed by a small number of effective parameters, whose values are determined self-consistently by fixed-point equations. Although non-rigorous in general, replica-theoretic predictions have been confirmed by rigorous approaches in a range of settings (Bayati & Montanari, 2011a;b) and have become a standard tool for characterizing the generalization error of high-dimensional regression problems under structured priors and penalties (Guo & Verdú, 2005; Rangan et al., 2009; Bereyhi & Müller, 2018; Bereyhi et al., 2019).

To apply this framework to our setting, we first reformulate the implicit bias derived in Theorem 4.1 as an explicit optimization problem:

$$\hat{\beta} := \arg\min_{\beta \in \mathbb{R}^D} \frac{1}{2\lambda} \|\mathbf{X}_{FT}\beta - \mathbf{y}_{FT}\|_2^2 + Q_k(\beta_{FT}), \quad (15)$$

where $\beta \in \mathbb{R}^D, k \in \mathbb{R}_+^D$. For $\lambda \to 0$, this precisely characterizes the implicit inductive bias of a diagonal linear network trained on the training data $(\mathbf{X}_{FT}, \mathbf{y}_{FT})$. We now characterize how accurately this optimization problem can recover the ground-truth vector $\beta^*_{FT} \in \mathbb{R}^D$, given the label

$$\mathbf{y}_{FT} = \mathbf{X}_{FT}\beta^*_{FT} + \varepsilon, \quad \varepsilon \sim \mathcal{N}(0, \sigma_0^2). \quad (16)$$

While our theory also applies to $\sigma_0^2 > 0$, we will focus on the noiseless case $\sigma_0^2 = 0$. We assume that $\beta^*_{FT}, \beta^*_{PT}$ are sampled from (8-10) and characterize the generalization error in the high-dimensional limit:

**Proposition 4.2.** *Let $N_{FT} \to \infty$ and $N_{FT}/D \to \alpha_{FT} > 0$. Then, under the replica assumption, the estimation problem* (15) *decouples into a scalar problem*

$$\hat{\beta}^{sc}(y; k, \theta) := \arg\min_{\beta \in \mathbb{R}} \left\{ \frac{(y - \beta)^2}{2\theta} + Q_k(\beta) \right\}. \quad (17)$$

*Specifically, $(\beta_d^*, \hat{\beta}_d, k_d)$ converge in distribution to*

$$\hat{\beta}_d = \hat{\beta}^{sc}(\beta_d^* + \eta; k_d, \theta), \quad \eta \sim \mathcal{N}(0, \theta_0). \quad (18)$$

*Here $\theta$ and $\theta_0$ are the "effective" regularization and noise parameters. They are not only governed by the external noise and regularization, but also by the noise and regularization induced from estimating the other parameters. These two properties are governed by the fixed-point equations*

$$p = \sum_{j=1}^{J} \pi_j \mathbb{E}_{\beta^*, k, \eta} \left[ (\hat{\beta}^{sc}(\beta^* + \eta; k, \theta) - \beta^*)^2 \right], \quad (19)$$

$$\chi = \theta \sum_{j=1}^{J} \pi_j \mathbb{E}_{\beta^*, k, \eta} \left[ \partial_y \hat{\beta}^{sc}(\beta^* + \eta; k, \theta) \right], \quad (20)$$

*where $\theta = \frac{\lambda + \chi}{\alpha}, \quad \theta_0 = \frac{\sigma_0^2 + p}{\alpha}$.*

*Proof.* The proposition follows from Proposition 1 in Bereyhi & Müller (2018), see Appendix C. □

This proposition, paired with Theorem 4.1, yields an exact formula for the generalization error associated with PT+FT in diagonal linear networks, as a function of weight initialization, task parameters, and data scale. As we will show in Section 5, different initialization choices induce distinct learning regimes, favoring different task structures.

# 5. Understanding Learning Regimes in PT+FT

The theoretical insights developed in Section 4 help us understand the inductive bias of PT+FT across the full spectrum of weight initialization and task parameters. We first use Theorem 4.1 to characterize how the initialization parameters influence the network's learning regime (Sections 5.1 and 5.2) and then use Proposition 4.2 to tie these different learning regimes to the task parameters for which they are favorable, both for perfect and imperfect pretraining (Section 5.3).

## 5.1. Learning Regimes in the Limit

Building on Theorem 4.1, we characterize the learning regime of the networks using two measures (introduced in Lippl & Lindsey (2024)):

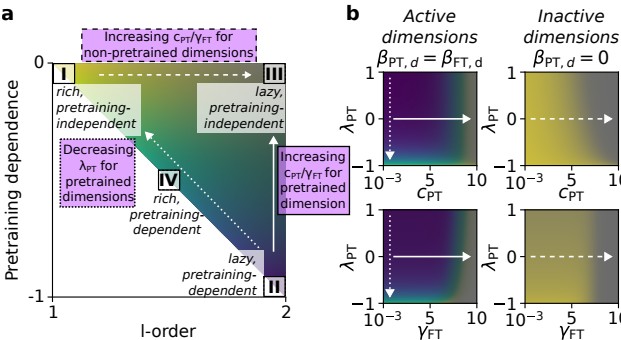

*Figure 2.* **Implicit bias and learning regimes of fine-tuning.** (a) $\ell$-order and pretraining dependence jointly define four different learning regimes in PT+FT. Different initialization parameters induce changes between these learning regimes, as indicated by the arrows. (b) We can interpolate between these four regimes by shifting the initialization parameters ($\hat{\beta}_{FT,d} = 1/\sqrt{\rho_{FT}}$, $\rho_{FT} = 0.1$). (For a color legend, see panel (a).) Crucial transitions, which we further highlight in the text and in panel (a), are indicated by the arrows. Simulation details can be found in Appendix E.

1. The $\ell$-order $:= \frac{\partial \log q_{k_d}(\beta_{FT,d})}{\partial \log |\beta_{FT,d}|}$, measures whether the network benefits from sparsity. $\ell$-order $= 1$ indicates a sparse inductive bias (e.g. the $\ell_1$-norm), whereas $\ell$-order $= 2$ corresponds to a non-sparse inductive bias (e.g. the $\ell_2$-norm).

2. The pretraining dependence[2], PD $:= \frac{\partial \log q_{k_d}(\beta_{FT,d})}{\partial \log |\beta_{PT,d}|}$ measures whether the network benefits from pretrained features. PD $= -1$ indicates that the penalty is inversely proportional to the magnitude of the pretrained feature $|\beta_{PT,d}|$, whereas PD $= 0$ indicates that the penalty does not depend on it at all.

We derive analytical formulas for these metrics as a function of both the initialization parameters and $\beta_{PT,d}$ and $\beta_{FT,d}$ (see Appendix B.2). We further identify a universal tension between $\ell$-order and PD:

**Proposition 5.1.** *Across the full spectrum of initialization, $\ell$-order $\in [1, 2]$, PD $\in [-1, 0]$, $\ell$-order + PD $\in [1, 2]$.*

*Proof.* We derive these ranges in Appendix B.2. They extend a finding from Lippl & Lindsey (2024) who found that $\ell$-order + PD $= 1$ for $c_{PT} \to 0, \lambda_{PT}, \gamma_{FT} = 0$. $\square$

In **Fig. 2a**, we identify three limiting regimes arising from the interplay between the $\ell$-order and pretraining dependence. Taking appropriate limits of the initialization parameters $\lambda_{PT}$ and $c_{PT}$ places the model in these regimes, each corresponding to a familiar norm-based regularization:

---

[2]Lippl & Lindsey (2024) introduced this metric as feature dependence; we call it pretraining dependence to emphasize that it characterizes the dependence on the pretraining function directly.

(I) A **rich, pretraining-independent regime**, (new dimensions learned; $\ell$-order $= 1$, PD $= 0$). This regime is governed by the $\ell_1$-norm: $Q(\beta_{FT}) \to \|\beta_{FT}\|_1$ and is induced e.g. in the limit $\lambda_{PT} \to -1$, $\gamma_{FT} \to 0$. We call an $\ell_1$-like inductive bias in diagonal linear networks "rich," because it arises from feature-learning dynamics (see Section 2).

(II) A **lazy, pretraining-dependent regime**, (reuse with minimal adaptation; $\ell$-order $= 2$, PD $= -1$). This regime is governed by the weighted $\ell_2$-norm: $Q(\beta_{FT}) \to \sum_{i=1}^{D} |\beta_{FT,i}|^2 / |\beta_{PT,i}|$. It is induced e.g. by considering $c_{PT} \to 0$ and $\lambda_{PT} \to 1$.

(III) A **lazy, pretraining-independent regime** ($\ell$-order $= 2$, PD $= 0$). This regime is governed by the unweighted $\ell_2$-norm: $Q(\beta_{FT}) \to \|\beta_{FT}\|_2$. It implies that fine-tuning behavior does not depend on pretraining and is induced e.g. by $c_{PT} \to \infty$ or $\gamma_{FT} \to \infty$.

We highlight the identification of the novel regime (III), which is accessible only in the fine-tuning setting and highlights that for badly chosen initialization parameters, pretraining will not create transferable features.

### 5.2. Full Phase Portrait of the Learning Regimes

In Section 5.1, our analysis was restricted to asymptotic behavior. However, in practice we may be operating in an intermediate learning regime where the parameters do not approach one of the above limits. Indeed, operating in such a regime will often confer a beneficial inductive bias: On the one hand, for pretraining to be useful, the penalty should depend on the active pretraining dimensions, i.e. ideally PD $= -1$. On the other hand, a sparse inductive bias substantially improves generalization performance (if the ground truth is also sparse), i.e. ideally $\ell$-order $= 1$. Proposition 5.1 highlights a fundamental tradeoff between these desiderata: it is impossible to achieve PD $= -1$ and $\ell$-order $= 1$. The set of possible values for the pair ($\ell$-order, $PD$) is given by a triangle whose edges are given by regimes (I-III) (**Fig. 2b**). We thus highlight an important intermediate regime:

(IV) The **rich, pretraining-dependent regime** ($\ell$-order $< 2$, $PD < 0$) achieves a balance between pretraining dependence and $\ell$-order, enabling the model to leverage both sparsity and feature reuse simultaneously.

Overall, the four learning regimes we highlight are distinguished by their pretraining dependence and sparsity bias/$\ell$-order. To understand how the initialization parameters impact the learning regime, we plot the full phase portrait of the learning regimes as a function of pretraining initialization (**Fig. 2b**), considering a typical value at the end of training,

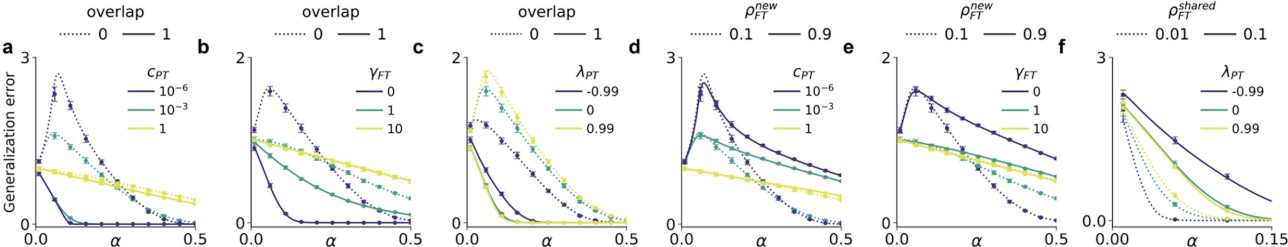

*Figure 3.* **Generalization curves for different initialization parameters and task parameters.** The generalization error $\mathcal{E}$ as a function of the data scale $\alpha_{FT}$. Lines depict replica predictions and points depict the results of our empirical simulations ($\pm 2$ standard errors). In all cases, $\rho_{PT} = 0.1$. We consider $c_{PT} = 10^{-3}$, $\lambda_{PT} = 0$, and $\gamma_{FT} = 0$, varying one initialization parameter for each panel. Simulation details can be found in Appendix E. (a-c) We consider either overlapping ($\rho_{FT}^{\text{shared}} = 0.1$, $\rho_{FT}^{\text{new}} = 0$) or distinct ($\rho_{FT}^{\text{shared}} = 0$, $\rho_{FT}^{\text{new}} = 0.1$) FT dimensions. (d,e) We consider $\rho_{FT}^{\text{shared}} = 0$ and vary $\rho_{FT}^{\text{new}}$. (f) We consider $\rho_{FT}^{\text{new}} = 0$ and vary $\rho_{FT}^{\text{shared}}$.

$\beta_{FT,d} = 1/\sqrt{\rho_{FT}}$ (setting $\rho_{FT} = 0.1$ as this will be a common setting in the next section). Note that because of our task generation process, we either have $\beta_{PT,d}^* = \pm 1/\sqrt{\rho_{PT}}$ (for active dimensions; we set $\rho_{PT} = 0.1$) or $\beta_{PT,d}^* = 0$ (for inactive dimensions). Plotting the phase diagram for both cases clarifies which learning regime we operate in for active and inactive dimensions.

For active dimensions where $\beta_{PT,d}^* = \pm 1/\sqrt{\rho_{PT}}$, changing the relative scale $\lambda_{PT}$ shifts the network from the *lazy, pretraining-dependent regime* (II) to the *rich, pretraining-independent regime* (I), with intermediate scales yielding the *rich, pretraining-dependent regime* (IV) (see the dotted line in **Fig. 2a,b**). Intuitively, a negative $\lambda_{PT}$ implies that the second-layer weights dominate over the first layer ($v > w$), while a positive $\lambda_{PT}$ indicates the opposite ($w > v$). When $\lambda_{PT}$ is large and negative, the first-layer representation is comparatively small, even after pretraining. After rescaling the second layer to $\gamma_{FT} = 0$, these large second-layer weights are reduced, decreasing the overall scale of the network—this drives the system toward the rich regime. In contrast, as $\lambda_{PT}$ increases, the learned first-layer representation becomes larger, driving the network into the *lazy, pretraining-dependent regime*. In contrast, increasing the absolute scale $c_{PT}$ or the fine-tuning re-initialization scale $\gamma_{FT}$ moves the system toward the *lazy, pretraining-independent regime* (III), where neither sparsity nor shared dimensions improve generalization (solid line in **Fig. 2a,b**).

For inactive dimensions ($\beta_{PT,d}^* = 0$), we find that PD = 0. Thus, instead of the two-dimensional continuum outlined above, we recover a one-dimensional continuum between the *lazy, pretraining-independent regime* (III) and the *rich, pretraining-independent regime* (I) (**Fig. 2b**, right column). Increasing $\gamma_{FT}$ or $c_{PT}$ shifts the inductive bias from regime (I) into regime (III) (see the dashed line in **Fig. 2a,b**). Additionally, (albeit less pronounced), increasing $\lambda_{PT}$ also shifts the inductive bias into a slightly lazier regime.

Overall, the different initialization parameters can therefore change the inductive bias of PT+FT along three axes: they

can control 1) whether the network benefits from shared dimensions with the pretraining task (measured by the PD for active dimensions), 2) whether the network benefits from sparsity in the new active dimensions (measured by the $\ell$-order for $\beta_{PT,d}^* = 0$), and 3) whether the network benefits from sparsity in the shared active dimensions (measured by the $\ell$-order for $\beta_{PT,d}^* > 0$). In the next section, we leverage Proposition 4.2 to study how these different initializations impact generalization performance across task parameters.

### 5.3. Learning Regime and Task Parameters Jointly Determine Generalization Error Curves

In practice, the implicit bias described above is reflected in the sample efficiency and generalization performance observed during fine-tuning. We solve the fixed-point equations derived in Proposition 4.2 to understand how the different learning regimes and task parameters impact generalization performance. In particular, we examine the impact of the different initialization parameters on the three axes outlined above. To validate our analytical results, we additionally directly train diagonal linear networks.

**Do We Benefit From Shared Dimensions?** If our penalty induces pretraining dependence, we should benefit from sharing dimensions between pretraining and fine-tuning. We therefore consider a fixed $\rho_{PT} = 0.1$ and compare the case $\rho_{FT}^{\text{shared}} = 0.1$, $\rho_{FT}^{\text{new}} = 0$ (in which case the pretraining and fine-tuning task share all their dimensions) to the case $\rho_{FT}^{\text{shared}} = 0$, $\rho_{FT}^{\text{new}} = 0.1$ (in which case they share no dimensions, but the overall sparsity level is matched). As $c_{PT}$ and $\gamma_{FT}$ decrease, performance becomes increasingly sensitive to task overlap (**Fig. 3a,b**). On the other hand, decreasing $\lambda_{PT}$ decreases the extent to which we benefit from shared dimensions (**Fig. 3c**). These trends are consistent with the phase-portrait analysis, which indicates that for pretrained dimensions, decreasing $\gamma_{FT}$ and $c_{PT}$ at fixed $\lambda_{PT}$ shifts the network from a *lazy, pretraining-independent regime* toward a *lazy, pretraining-dependent regime* (solid line in **Fig. 2**). In contrast, decreasing $\lambda_{PT}$ shifts the model from

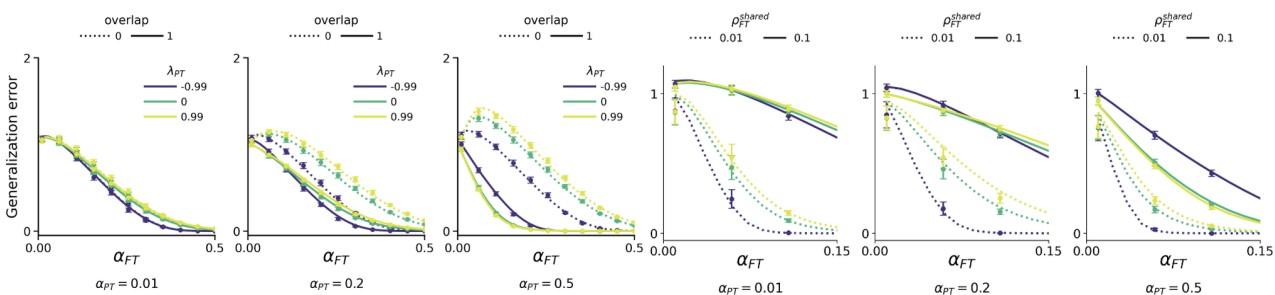

*Figure 4.* **Effect of pretraining strength ($\alpha_{PT}$) on the role of relative scale ($\lambda_{PT}$).** Generalization error as a function of fine-tuning data scale $\alpha_{FT}$ for different values of $\alpha_{PT}$ and task overlap. For small $\alpha_{PT}$, behavior is largely insensitive to $\lambda_{PT}$. As $\alpha_{PT}$ increases, the effect of $\lambda_{PT}$ becomes stronger, particularly in the no-overlap setting. For $\alpha_{PT} \geq 1$, all choices of $\lambda_{PT}$ perform similarly under full overlap. Lines depict replica predictions; points depict empirical simulations ($\pm 2$ standard errors).

a *lazy, pretraining-dependent regime* to a *rich, pretraining-independent regime* (dotted line in **Fig. 2**).

**Do We Benefit From Sparsity In New Active Dimensions?** For inactive dimensions (i.e. $\beta^*_{PT,d} = 0$), we can either be in a rich regime where we benefit from a sparse set of new dimensions, or a lazy regime where we learn sparse and non-sparse sets equally well. We therefore consider a task with no shared dimensions with the pretraining task ($\rho_{PT} = 0.1, \rho^{\text{shared}}_{FT} = 0$) and compare performance across different levels of sparsity on the fine-tuning task ($\rho_{FT} = 0.1, 0.9$). We find that as $c_{PT}$ and $\gamma_{FT}$ decrease, sparsity becomes increasingly beneficial for performance (**Fig. 3d,e**). This observation aligns with the phase-portrait analysis, which shows that for $\beta^*_{PT,d} = 0$ the model transitions from a *lazy, pretraining-independent regime*, which has no sparsity bias, to a *rich, pretraining-independent regime*, which has a sparsity bias (dashed line in **Fig. 2**).

**Do We Benefit From Sparsity In Shared Active Dimensions?** Rich learning regimes should also benefit from sparsity in pretrained dimensions (with $\beta^*_{PT,d} = \pm 1/\sqrt{\rho_{PT}}$). We therefore consider fully overlapping dimensions ($\rho_{PT} = 0.1, \rho^{\text{new}}_{FT} = 0$) and compare $\rho^{\text{shared}}_{FT} = 0.1$ to $\rho^{\text{shared}}_{FT} = 0.01$. We observe that as $\lambda_{PT}$ decreases, the network increasingly benefits from sparsity (**Fig. 3f**). However, this effect vanishes if $c_{PT}$ or $\gamma_{FT}$ are very large. This observation again aligns with the phase-portrait analysis: as $\lambda_{PT}$ decreases, we move into the *rich, pretraining-dependent regime* (dotted line in **Fig. 2**), but if $c_{PT}$ or $\gamma_{FT}$ increase, we move into the *lazy, pretraining-independent regime* regardless of $\lambda_{PT}$ (solid line in **Fig. 2**).

**The Role of Pretraining Strength.** So far, we have focused on the case $\alpha_{PT} \geq 1$, in which pretraining perfectly recovers the ground truth. In practice, however, pretraining is often imperfect. We therefore extend our analysis to $\alpha_{PT} < 1$, where $\hat{\beta}_{PT}$ deviates from $\beta^*_{PT}$.

Intuitively, $\alpha_{PT}$ controls the accuracy of the active dimen-

sions learned during pretraining, while $\lambda_{PT}$ continues to determine whether these dimensions are reused or adapted during fine-tuning. In **Fig. 4**, we vary $\alpha_{PT}$ across different levels of task overlap. For small $\alpha_{PT}$, behavior is largely insensitive to $\lambda_{PT}$, as the pretraining signal is too weak to be reused. As $\alpha_{PT}$ increases, the effect of $\lambda_{PT}$ becomes progressively stronger, particularly in the no-overlap setting where fine-tuning must learn new features. On the other hand, while for $\alpha_{PT} \geq 1$ negative $\lambda_{PT}$ makes generalization worse under full overlap (**Fig. 3f**), for $\alpha_{PT} < 1$ it can improve performance even in that case, as it allows the model to refine the imperfectly learned pretraining structure. This highlights that imbalanced initialization may be especially important when pretraining data is limited.

**Summary.** Taken together, the generalization curves elucidate how initialization parameters interact with task parameters to shape generalization behavior. In particular, the generalization curves demonstrate the same trade-off between sparsity and pretraining dependence we predicted from Proposition 5.1: as $\lambda_{PT}$ becomes more negative, we move along the diagonal representing the trade-off between $\ell$-order and pretraining dependence (dotted line in **Fig. 2**). As a result, the network benefits less from overlap: for $\rho^{\text{shared}}_{FT} = 0.1$ (i.e. full overlap), decreasing $\lambda_{PT}$ makes generalization worse (**Fig. 3f**). At the same time, the network benefits more from sparsity: for $\rho^{\text{shared}}_{FT} = 0.01$, trading off a stronger sparsity bias for a weaker pretraining dependence is worthwhile, and a smaller $\lambda_{PT}$ yields better performance. Overall, this identifies $\lambda_{PT}$ as a crucial factor for changing the inductive bias of PT+FT and highlights that the optimal learning regime will depend on the task parameters (**Fig. 6**).

## 6. Large-Scale Vision Experiments

Our theoretical analysis and small-scale experiments suggest that modifying pretraining and re-initialization parameters provides a theoretically principled mechanism for inducing useful feature learning during fine-tuning. To evaluate

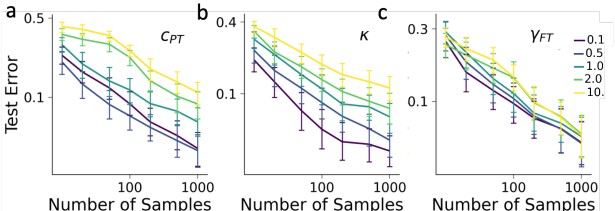

*Figure 5.* **ResNet CIFAR-100.** Generalization performance as a function of the number of samples and initialization parameters. We vary (a) the absolute scale of initialization by multiplying all weights in the network by $c_{PT}$, (b) the relative scale of initialization by multiplying the first three blocks of the ResNet by $\kappa$ (equivalent to $\lambda_{PT}$), and (c) the readout initialization by multiplying the readout by $\gamma_{FT}$ (Simulation details: Appendix E).

this prediction, we pretrain a ResNet on 98 classes from CIFAR-100 and subsequently fine-tune it on the remaining two classes. The class split is sampled randomly and repeated fifty times. We design three experimental settings that translate the theoretical parameters $\lambda_{PT}$, $c_{PT}$, and $\gamma_{FT}$ to a realistic deep-network architecture. The ResNet-18 architecture consists of an initial embedding layer followed by four stages of residual blocks. To induce imbalance across network layers, we upscale the embedding layer and the first three residual stages by a constant factor $\kappa$. To implement an equivalent notion of $c_{PT}$ scaling in this architecture, we scale all network parameters by a constant factor. Finally, to implement an analogue of $\gamma_{FT}$, we rescale the parameters of the final classification layer after pretraining[3]. Further experimental details are provided in Appendix D.

In **Fig. 5a**, we observe that a ResNet initialized with a non-standard small value of $\kappa$—where $\kappa < 1$ corresponds to a negative relative scaling—exhibits improved generalization during fine-tuning. We note that the ordering between individual $\kappa$ values is not always consistent (e.g. the blue and violet lines in **Fig. 5a**), though these differences are not statistically significant. This may reflect a trade-off between initialization scale and optimizer dynamics, potentially compounded by the fact that sparsity and overlap are not explicitly controlled in this setting. Similarly, as shown in **Fig. 5b**, increasing the scale of $c_{PT}$ also leads to degraded fine-tuning generalization, in agreement with our theoretical predictions. Finally, **Fig. 5c** shows that decreasing the scale of $\gamma_{FT}$ improves fine-tuning generalization for intermediate sample sizes. Overall, all parameters identified by the theory have a meaningful impact on generalization during fine-tuning in practice.

To verify that these interventions induce the *rich, pretraining-dependent regime* predicted by our theory, we conduct a representation-level analysis. We measure the

dimensionality of the network representation before and after fine-tuning using the participation ratio (PR; Gao et al., 2017), and the number of dimensions shared between the two states using the effective number of shared dimensions (ENSD; Giaffar et al., 2024). The pretraining-dependent rich regime predicts that fine-tuning should compress the representation, i.e. $\mathrm{PR}(X_{FT}) < \mathrm{PR}(X_{PT})$, while preserving its alignment with the pretrained representation, i.e. $\mathrm{ENSD}(X_{PT}, X_{FT}) \approx \mathrm{PR}(X_{FT})$. In Appendix D, we show that all three interventions ($\kappa$, $c_{PT}$, $\gamma_{FT}$, as well as the $c_{FT}$ rescaling heuristic of Lippl & Lindsey (2024)) produce this signature in the final ResNet layer. We further investigate the robustness of our results by fine-tuning on a different dataset (SVHN; we use this to probe the impact of decreased task similarity, Appendix D.5), running experiments with different optimizers and data splits (Appendices D.7 and D.6), and exploring the impact of balancedness on Transformers pretrained and fine-tuned on modular addition (Appendix D.8).

Several limitations merit discussion when interpreting these results. Our theoretical analysis is developed in a diagonal two-layer setting, while ResNets are nonlinear and include residual connections and batch normalization. Our experiments also do not explicitly control sparsity levels or feature overlap. We leave a more careful study of these parameters in practical networks to future work.

## 7. Conclusion

Despite the prevalence of the pretraining–fine-tuning (PT+FT) pipeline in modern deep learning, a comprehensive understanding of the inductive bias of PT+FT, and how it relates to initialization structure and task parameters, has remained elusive. Here we developed an end-to-end theory of PT+FT in diagonal linear networks, analytically computing the generalization error as a function of initialization parameters, task parameters, and data scale. Albeit based on a simplified neural network model, our analysis provides quantitative insights relevant to practical scenarios in machine learning and neuroscience. In particular, it underscores the importance of relative weight scales across layers. Our work also opens up several directions for future investigation. A similar analysis could be conducted in a continual learning setting, allowing us to characterize the conditions on task similarity and initialization that support forward and backward transfer. Extending the framework to nonlinear neural networks would allow it to capture richer forms of feature overlap and could help uncover how feature reuse and adaptation arise across both artificial and biological systems, e.g. via influence functions (Koh & Liang, 2017). Finally, the implicit inductive biases identified here could inform the design of explicit regularization objectives that drive networks into the desired learning regimes.

---

[3]For completeness, we report in Appendix D the heuristic proposed by Lippl & Lindsey (2024)

## Acknowledgments

NA thanks the Rafael del Pino Foundation for financial support. This research was funded in whole or in part by the Austrian Science Fund (FWF) 10.55776/COE12. SL was supported by a grant from the Simons Foundation International 542939SPI, LFA; SFI-AN-NC-GB-Culmination-00003215-01. This work was supported by a Schmidt Science Polymath Award to AS, and the Sainsbury Wellcome Centre Core Grant from Wellcome (219627/Z/19/Z) and the Gatsby Charitable Foundation (GAT3850). FM is funded by a MRC Career Development Award (MR/T010614/1), a UKRI Advanced Pain Discovery Platform grant (MR/W027593/1), and a EPSRC/MRC Programme Grant (UKRI1970).

## Impact Statement

This paper presents work whose goal is to advance the field of Machine Learning. There are many potential societal consequences of our work, none of which we feel must be specifically highlighted here.

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

# A. Extended Results

We showcase the four limit regimes by plotting the generalization curves across three different ground truth structures, varying sparsity and overlap:

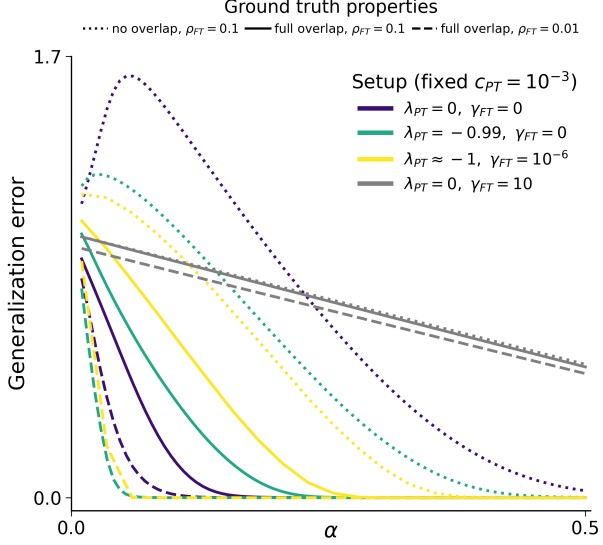

*Figure 6.* **Comparison of the four fine-tuning learning regimes.** We show three task parameter settings: 1) no overlap between pretraining and fine-tuning dimensions; 2) identical pretraining and fine-tuning dimensions; 3) fine-tuning dimension as a subset of pretraining dimensions. We should for initialization parameter settings: a lazy, pretraining-dependent regime (II, shown in purple), a lazy, pretraining-independent regime (III, shown in grey), an intermediate rich, pretraining-dependent regime (IV, shown in green), and a richer, less pretraining-dependent regime (I, shown in yellow). We observe that for the task parameter setting without any overlap, the regime approaching the rich, pretraining-independent regime (in yellow) is optimal. For the complete overlap between pretraining and fine-tuning dimensions, the lazy, pretraining-dependent regime (II, in purple) is optimal. Finally, for the task where the fine-tuning dimensions are a subset of pretraining dimensions, the rich, pretraining-dependent regime (IV, in green) is optimal. All of these observations are predicted by our theoretical insight in the inductive bias of PT+FT.

# B. Theoretical Analysis of the PT+FT Learning Regimes

## B.1. Implicit Bias

**Theorem 4.1** (Implicit bias). *Consider a diagonal linear network trained on PT+FT (as described in Section 3.2). Then, the gradient flow solution at convergence is given by*

$$\arg\min_{\beta_{FT}} Q_k(\beta_{FT}) \quad s.t. \ \mathbf{X}_{FT}^{\top}\beta_{FT} = \mathbf{y}_{FT}, \tag{11}$$

$$where \quad Q_k(\beta_{FT}) = \sum_{d=1}^{D} q_{k_d}(\beta_{FT,d}), \tag{12}$$

$$q_k(z) = \frac{\sqrt{k}}{4}\left(1 - \sqrt{1 + \frac{4z^2}{k}} + \frac{2z}{\sqrt{k}}\operatorname{arcsinh}\left(\frac{2z}{\sqrt{k}}\right)\right), \tag{13}$$

$$k_d = \left(2c_{PT}(1+\lambda_{PT})\left(1 + \sqrt{1 + (\hat{\beta}_{PT,d}/c_{PT})^2}\right) + \gamma^2\right)^2. \tag{14}$$

*Proof.* We follow a similar proof method as in (Azulay et al., 2021). We assume the pretraining quantities:

$$\mathbf{w}^{+}(0)^2 - \mathbf{v}^{+}(0)^2 = \tilde{\lambda}_{PT}$$
$$\mathbf{w}^{-}(0)^2 - \mathbf{v}^{-}(0)^2 = \tilde{\lambda}_{PT}$$
$$\mathbf{w}^{+}(0)\mathbf{w}^{-}(0) + \mathbf{v}^{+}(0)\mathbf{v}^{-}(0) = c_{PT},$$

where $\lambda_{PT} = \tilde{\lambda}_{PT}/c_{PT}$.

Reparameterize:

$$\mathbf{w}^+ = \sqrt{\tilde{\lambda}_{PT}} \cosh \theta^+, \quad \mathbf{w}^- = \sqrt{\tilde{\lambda}_{PT}} \cosh \theta^-$$

$$\mathbf{v}^+ = \sqrt{\tilde{\lambda}_{PT}} \sinh \theta^+, \quad \mathbf{v}^- = \sqrt{\tilde{\lambda}_{PT}} \sinh \theta^-$$

We use the conserved quantities $c_{PT}, \tilde{\lambda}_{PT}$ to get an expression for $\theta^+$ and $\theta^-$.

$$\cosh \theta^+ \cosh \theta^- + \sinh \theta^+ \sinh \theta^- = \frac{c_{PT}}{\tilde{\lambda}_{PT}}$$

$$\cosh \theta^+ \sinh \theta^+ - \cosh \theta^- \sinh \theta^- = \frac{\beta_{PT}}{\tilde{\lambda}_{PT}}$$

Therefore

$$\mathbf{w}^+ = \sqrt{\tilde{\lambda}_{PT}} \cosh \left( \frac{1}{2} \left[ \mathrm{arcosh} \left( \frac{c_{PT}}{\tilde{\lambda}_{PT}} \right) + \mathrm{arsinh} \left( \frac{\beta_{PT}}{c_{PT}} \right) \right] \right)$$

$$\mathbf{v}^+ = \sqrt{\tilde{\lambda}_{PT}} \sinh \left( \frac{1}{2} \left[ \mathrm{arcosh} \left( \frac{c_{PT}}{\tilde{\lambda}_{PT}} \right) + \mathrm{arsinh} \left( \frac{\beta_{PT}}{c_{PT}} \right) \right] \right)$$

$$\mathbf{w}^- = \sqrt{\tilde{\lambda}_{PT}} \cosh \left( \frac{1}{2} \left[ \mathrm{arcosh} \left( \frac{c_{PT}}{\tilde{\lambda}_{PT}} \right) - \mathrm{arsinh} \left( \frac{\beta_{PT}}{c_{PT}} \right) \right] \right)$$

$$\mathbf{v}^- = \sqrt{\tilde{\lambda}_{PT}} \sinh \left( \frac{1}{2} \left[ \mathrm{arcosh} \left( \frac{c_{PT}}{\tilde{\lambda}_{PT}} \right) - \mathrm{arsinh} \left( \frac{\beta_{PT}}{c_{PT}} \right) \right] \right)$$

After pretraining, we reinitialize parameters:

$$\mathbf{w}_{FT}^+(0) = \mathbf{w}_{FT}^-(0) = \mathbf{w}_{PT}^+(\infty) + \mathbf{w}_{PT}^-(\infty)$$

$$\mathbf{v}_{FT}^+(0) = \mathbf{v}_{FT}^-(0) = \gamma_{FT}$$

We are interested in the initial conserved quantity before finetuning, $c_{FT}$.

By definition:

$$
\begin{aligned}
c_{FT} &= \mathbf{w}_{FT}^+(0)\mathbf{w}_{FT}^-(0) + \mathbf{v}_{FT}^+(0)\mathbf{v}_{FT}^-(0) \\
&= \left( \mathbf{w}_{PT}^+(\infty) + \mathbf{w}_{PT}^-(\infty) \right)^2 + \gamma_{FT}^2 \\
&= \left( \mathbf{w}_{PT}^+(\infty) \right)^2 + 2\,\mathbf{w}_{PT}^+(\infty)\,\mathbf{w}_{PT}^-(\infty) + \left( \mathbf{w}_{PT}^-(\infty) \right)^2 + \gamma_{FT}^2
\end{aligned}
$$

Let

$$A = \mathrm{arcosh} \left( \frac{c_{PT}}{\tilde{\lambda}_{PT}} \right), \qquad B = \mathrm{arsinh} \left( \frac{\beta_{PT}}{c_{PT}} \right)$$

We compute:

$$\mathbf{w}_{PT}^+(\infty)^2 = \tilde{\lambda}_{PT} \cosh^2 \left( \frac{1}{2}(A + B) \right) = \tilde{\lambda}_{PT} \left( \frac{1 + \cosh(A + B)}{2} \right)$$

$$\mathbf{w}_{PT}^-(\infty)^2 = \tilde{\lambda}_{PT} \cosh^2 \left( \frac{1}{2}(A - B) \right) = \tilde{\lambda}_{PT} \left( \frac{1 + \cosh(A - B)}{2} \right)$$

So:

$$\mathbf{w}_{PT}^+(\infty)^2 + \mathbf{w}_{PT}^-(\infty)^2 = \tilde{\lambda}_{PT} \left( 1 + \frac{\cosh(A + B) + \cosh(A - B)}{2} \right)$$

Recall the identity:

$$\cosh(A + B) + \cosh(A - B) = 2\cosh A \cosh B$$

Therefore:

$$\mathbf{w}_{PT}^{+}(\infty)^2 + \mathbf{w}_{PT}^{-}(\infty)^2 = \tilde{\lambda}_{PT}\left(1 + \cosh A \cosh B\right)$$

Now:

$$2\mathbf{w}_{PT}^{+}(\infty)\mathbf{w}_{PT}^{-}(\infty) = 2\sqrt{\tilde{\lambda}_{PT}}\cosh\left(\frac{A+B}{2}\right)\cdot\sqrt{\tilde{\lambda}_{PT}}\cosh\left(\frac{A-B}{2}\right)$$

$$= 2\tilde{\lambda}_{PT}\cosh\left(\frac{A+B}{2}\right)\cosh\left(\frac{A-B}{2}\right)$$

$$= \tilde{\lambda}_{PT}\left(\cosh A + \cosh B\right),$$

using the identity $2\cosh x \cosh y = \cosh(x + y) + \cosh(x - y)$.

Now compute:

$$\cosh A = \frac{c_{PT}}{\tilde{\lambda}_{PT}}, \qquad \cosh B = \cosh\left(\mathrm{asinh}\left(\frac{\beta^{aux}}{c_{PT}}\right)\right) = \sqrt{1 + \left(\frac{\beta^{aux}}{c_{PT}}\right)^2}$$

$$c_{\mathrm{FT}} = \tilde{\lambda}_{\mathrm{PT}}\frac{c_{\mathrm{PT}}}{\tilde{\lambda}_{\mathrm{PT}}}\sqrt{1 + \left(\frac{\beta_{PT}}{c_{\mathrm{PT}}}\right)^2}$$

$$+ \tilde{\lambda}_{\mathrm{PT}}\left(\frac{c_{\mathrm{PT}}}{\tilde{\lambda}_{\mathrm{PT}}} + \sqrt{1 + \left(\frac{\beta_{PT}}{c_{\mathrm{PT}}}\right)^2}\right) + \gamma_{FT}^2$$

$$c_{\mathrm{FT}} = \left(\tilde{\lambda}_{\mathrm{PT}} + c_{\mathrm{PT}}\right)\left(1 + \sqrt{1 + \left(\frac{\beta_{PT}}{c_{\mathrm{PT}}}\right)^2}\right) + \gamma_{FT}^2$$

Now, from Azulay et al. (2021) we know that the implicit bias of a diagonal linear network is parameterized as in Eq. (1) that converges to a zero loss solution can be expressed as:

$$\beta_*(\infty) = \arg\min_{\beta_{FT}} Q_k(\beta_{FT}) \quad \text{s.t.} \quad \mathbf{X}_{FT}^{\top}\beta_{FT} = \mathbf{y}_{FT},$$

$$\text{where} \quad Q_k(\beta_{FT}) = \sum_{i=1}^{D} q_{k_i}(\beta_{FT,i}),$$

$$\text{with} \quad q_k(z) = \frac{\sqrt{k}}{4}\left(1 - \sqrt{1 + \frac{4z^2}{k}} + \frac{2z}{\sqrt{k}}\mathrm{arcsinh}\left(\frac{2z}{\sqrt{k}}\right)\right),$$

where $k_i = 4c_i^2$. In our case $k_i = 4c_{FT,i}^2 = \left(2(\tilde{\lambda}_{\mathrm{PT,i}} + c_{\mathrm{PT,i}})\left(1 + \sqrt{1 + (\beta_{\mathrm{PT,i}}/c_{\mathrm{PT,i}})^2}\right) + \gamma_{FT}^2\right)^2$. Hence $k_i = 4c_{FT,i}^2$.

The theorem follows from replacing $\tilde{\lambda}_{\mathrm{PT},i} = \lambda_{PT,i}c_{PT,i}$. $\qquad\square$

## B.2. Pretraining Importance and $\ell$-order

We now derive an analytical expression for $\ell$-order and pretraining dependence (PD), where

$$\ell\text{-order} := \frac{\partial \log q_k(\beta_{FT})}{\partial \log \beta_{FT}} \quad \text{and} \quad PD := \frac{\partial \log q_k(\beta_{FT})}{\partial \log \beta_{PT}}. \tag{21}$$

We define

$$\zeta := \frac{2\beta_{FT}}{\sqrt{k}}, \quad s := \frac{\beta_{PT}}{c_{PT}}, \quad \phi(z) = 1 - \sqrt{1+z^2} + z \operatorname{arcsinh}(z). \tag{22}$$

We additionally define the quantity

$$\kappa := \frac{\partial \log k}{\partial \log \beta_{PT}}. \tag{23}$$

Intuitively, $\kappa$ captures how much a certain function that was learned during pretraining transfers to fine-tuning features.

**Proposition B.1.** *We find that*

$$\ell\text{-order} = \frac{\zeta \operatorname{asinh}(\zeta)}{\phi(\zeta)}, \tag{24}$$

$$\kappa = \frac{4c_{PT}(1+\lambda_{PT})s^2}{(2c_{PT}(1+\lambda_{PT})(1+\sqrt{1+s^2}) + \gamma_{FT}^2)\sqrt{1+s^2}}, \tag{25}$$

$$PD = \frac{1 - \ell\text{-order}}{2}\kappa. \tag{26}$$

*Notably,*

$$\ell\text{-order} \in [1,2], \quad \kappa \in [0,2), \quad PD \in [-1,0], \tag{27}$$

*and*

$$\ell\text{-order} + PD \in [1,2] \tag{28}$$

*Proof.* First, with $\zeta := 2\beta_{FT}/\sqrt{k}$ we can rewrite

$$q_k(\beta_{FT}) = \frac{\sqrt{k}}{4}\left(1 - \sqrt{1+\zeta^2} + \zeta \operatorname{arcsinh}(\zeta)\right) = \frac{\sqrt{k}}{4}\phi(\zeta).$$

We now first derive the closed forms before deriving bounds for the different quantities.

**Closed form for $\ell$-order.** For fixed $k$,

$$\ell\text{-order} = \frac{\partial \log q_k(\beta_{FT})}{\partial \log \beta_{FT}} = \frac{\partial \log \phi(\zeta)}{\partial \log \zeta} = \zeta \frac{\phi'(\zeta)}{\phi(\zeta)}.$$

Since $\phi'(\zeta) = \operatorname{asinh}(\zeta)$,

$$\ell\text{-order} = \frac{\zeta \operatorname{asinh}(\zeta)}{\phi(\zeta)}.$$

**Closed form for $\kappa$.** Write $\sqrt{k} = A$ where

$$A := 2(\hat{\lambda}_{PT} + c_{PT})(1 + \sqrt{1+s^2}) + \gamma_{FT}^2, \quad s := \beta_{PT}/c_{PT}.$$

Then $k = A^2$, so

$$\kappa = \frac{\partial \log k}{\partial \log \beta_{PT}} = 2\frac{\partial \log A}{\partial \log \beta_{PT}}.$$

Since $s = \beta_{PT}/c_{PT}$ we have $\partial \log s/\partial \log \beta_{PT} = 1$. Also,

$$\frac{d}{ds}\sqrt{1+s^2} = \frac{s}{\sqrt{1+s^2}}, \quad \Rightarrow \quad \frac{\partial A}{\partial s} = 2(\hat{\lambda}_{PT} + c_{PT})\frac{s}{\sqrt{1+s^2}}.$$

Thus

$$\frac{\partial \log A}{\partial \log \beta_{PT}} = \frac{s}{A}\frac{\partial A}{\partial s} = \frac{s}{A} \cdot 2(\hat{\lambda}_{PT} + c_{PT})\frac{s}{\sqrt{1+s^2}} = \frac{2(\hat{\lambda}_{PT} + c_{PT})s^2}{A\sqrt{1+s^2}},$$

and therefore

$$\kappa = \frac{4(\hat{\lambda}_{PT} + c_{PT})s^2}{A\sqrt{1+s^2}} = \frac{4(\hat{\lambda}_{PT} + c_{PT})\,s^2}{\left(2(\hat{\lambda}_{PT} + c_{PT})(1+\sqrt{1+s^2}) + \gamma_{FT}^2\right)\sqrt{1+s^2}}.$$

**Closed form for $PD$.** By definition,

$$PD = \frac{\partial \log q_k(\beta_{FT})}{\partial \log \beta_{PT}} = \frac{\partial \log q_k(\beta_{FT})}{\partial \log k} \cdot \frac{\partial \log k}{\partial \log \beta_{PT}} = \frac{\partial \log q_k(\beta_{FT})}{\partial \log k} \cdot \kappa.$$

Now

$$\log q_k(\beta_{FT}) = \frac{1}{2}\log k + \log \phi(\zeta) - \log 4,$$

and $\frac{\partial \log \zeta}{\partial \log k} = -1/2$. Hence

$$\frac{\partial \log q_k(\beta_{FT})}{\partial \log k} = \frac{1}{2} + \frac{\partial \log \phi(\zeta)}{\partial \log \zeta} \cdot \frac{\partial \log \zeta}{\partial \log k} = \frac{1}{2} - \frac{1}{2}\,\ell\text{-order},$$

which yields

$$PD = \left(\frac{1}{2} - \frac{1}{2}\,\ell\text{-order}\right)\kappa = \frac{1 - \ell\text{-order}}{2}\kappa.$$

**Bounding $\ell$-order.** Assume $\zeta \geq 0$. First, since $1 - \sqrt{1+\zeta^2} \leq 0$,

$$\phi(\zeta) \leq \zeta \operatorname{arcsinh}(\zeta) \quad \Rightarrow \quad \ell\text{-order} = \frac{\zeta \operatorname{arcsinh}(\zeta)}{\phi(\zeta)} \geq 1.$$

For the upper bound, we show $\phi(\zeta) \geq \frac{1}{2}\zeta \operatorname{arcsinh}(\zeta)$, i.e.

$$\zeta \operatorname{arcsinh}(\zeta) \geq 2(\sqrt{1+\zeta^2} - 1).$$

Let $\zeta = \sinh t$ with $t = \operatorname{arcsinh}(\zeta) \geq 0$, so $\sqrt{1+\zeta^2} = \cosh t$. The inequality becomes

$$t \sinh t \geq 2(\cosh t - 1).$$

Define $h(t) := t \sinh t - 2(\cosh t - 1)$. Then $h(0) = 0$,

$$h'(t) = t \cosh t - \sinh t, \qquad h''(t) = t \sinh t \geq 0 \quad (t \geq 0).$$

Thus $h'$ is increasing and $h'(0) = 0$, so $h'(t) \geq 0$ for $t \geq 0$, implying $h(t) \geq 0$. Hence $\ell$-order $\leq 2$.

**Bounding $\kappa$.** Clearly $\kappa \geq 0$. Also $A \geq 2(\hat{\lambda}_{PT} + c_{PT})(1 + \sqrt{1+s^2})$, so

$$\kappa \leq \frac{4(\hat{\lambda}_{PT} + c_{PT})s^2}{2(\hat{\lambda}_{PT} + c_{PT})(1+\sqrt{1+s^2})\sqrt{1+s^2}} = 2 \cdot \frac{s^2}{\sqrt{1+s^2}\,(1+\sqrt{1+s^2})}.$$

Let $B := \sqrt{1+s^2} \geq 1$. Since $s^2 = B^2 - 1$,

$$2 \cdot \frac{B^2 - 1}{B(1+B)} = 2 \cdot \frac{B-1}{B} < 2,$$

with equality approached only in the limit $B \to \infty$ and $\gamma_{FT} \to 0$. Hence $\kappa \in [0, 2)$.

**Bounding $PD$ and $\ell$-order+$PD$.** From $PD = \frac{1-\ell\text{-order}}{2}\kappa$, the bounds $\ell$-order $\in [1,2]$ and $\kappa \in [0,2)$ imply

$$PD \leq 0, \qquad PD \geq \frac{1-2}{2} \cdot 2 = -1,$$

so $PD \in [-1, 0]$. Finally,

$$\ell\text{-order} + PD = \ell\text{-order} + \frac{1-\ell\text{-order}}{2}\kappa = \left(1 - \frac{\kappa}{2}\right)\ell\text{-order} + \frac{\kappa}{2}.$$

Since $\kappa/2 \in [0,1)$, this is a convex combination of $\ell$-order $\in [1,2]$ and 1, hence it lies in $[1,2]$. $\qquad\square$

## C. Replica Theory

### C.1. Proof of Proposition 4.2

We considering following generative model for $(\beta^*_{PT,d}, \beta_{FT,d})$.

**Definition C.1.** We assume that there are $J$ underlying groups. Each group is sampled with probability $\pi_j$, $\sum_{j=1}^{J} \pi_j = 1$, denoted by $j \sim \text{Cat}(\pi)$. Each group has associated pretraining and fine-tuning distributions $p_{PT}^{(j)}$ and $p_{FT}^{(j)}$ which are independent when conditioned on $j$:

$$j \sim \text{Cat}(\pi), \; \beta^*_{PT,d} \sim p_{PT}^{(j)}, \; \beta^*_{FT,d} \sim p_{FT}^{(j)}. \tag{29}$$

This means that any dependence between $\beta_{PT}$ and $\beta_{FT}$ is mediated by their respective group membership $j$. We apply Proposition 1 of Bereyhi & Müller (2018) to the estimator (15).

**Step 1: Notation.** We begin by noting a few notational differences to their setting. Specifically, to translate our setting into theirs, we set

$$A := X, \quad x := \beta, \quad c_d := k_d, \quad u_j(v; c) := q_c(v). \tag{30}$$

Moreover,

$$g_j^{\text{dec}} \equiv \hat{\beta}^{\text{sc}}, \tag{31}$$

except that we write out the dependence on $\theta$ explicitly. Finally, they treat $(c_d)$ as a deterministic sequence, whereas we consider a particular probability distribution for each block, $p_c^{(j)}$. However, because they average over coordinates, we know that this converges to the corresponding block-mixture expectations, weighted by $\pi_j = \lim_{D \to \infty} |B_j|/D$

**Step 2: Simplifying for i.i.d. X.** Proposition 1 of Bereyhi & Müller (2018) states (under the replica assumption) that, in the high-dimensional limit, the joint law of a typical coordinate decouples into a scalar Gaussian channel: for $d \in B_j$,

$$(\beta^*_d, \hat{\beta}_d, k_d) \; \Rightarrow \; (\beta^*, \hat{\beta}^{\text{sc}}(\beta^* + \eta; k, \theta), k), \qquad \eta \sim \mathcal{N}(0, \theta_0),$$

where $(\beta^*, k) \sim p_\beta^{(j)} \otimes p_k^{(j)}$ and the block is drawn according to $\pi_j$. Moreover, the scalar estimator is the MAP operator

$$\hat{\beta}^{\text{sc}}(y; k, \theta) := \arg\min_{\beta \in \mathbb{R}} \left\{ \frac{(y - \beta)^2}{2\theta} + Q_k(\beta) \right\},$$

which matches our definition.

**Step 3: Simplifying the fixed-point equations.** We now consider their resulting fixed-point equations and show how they evaluate to the fixed-point equations presented in our proposition. First, we note that because $X$ is i.i.d. with variance $1/D$, we can evaluate the R-transform. Specifically, $X$ has variance $1/D$. Thus $\frac{1}{\alpha}X$ has variance $1/N$. Thus, $R_{\frac{1}{\alpha}X}(\omega) = \frac{\alpha}{\alpha-\omega}$. By the properties of the R-transform (see e.g. Müller et al. (2013)),

$$R_X(\omega) = \alpha R_{\frac{1}{\alpha}X}(\alpha\omega) = \frac{\alpha}{1-\omega}. \tag{32}$$

Thus,

$$\theta = \frac{\lambda + \chi}{\alpha}, \quad \theta_0 = \frac{\sigma_0^2 + p}{\alpha}. \tag{33}$$

For the fixed-point equation for $\chi$, we additionally simplify the expression a little bit:

$$\frac{\theta_0}{\theta}\chi = \sum_{j=1}^{J} \pi_j \mathbb{E}_{\beta^*, k, \eta}\left[(\hat{\beta}^{\mathrm{sc}}(\beta^* + \eta; k, \theta) - \beta^*)\eta\right] \overset{(1)}{=} \sum_{j=1}^{J} \pi_j \mathbb{E}_{\beta^*, k, \eta}\left[\hat{\beta}^{\mathrm{sc}}(\beta^* + \eta; k, \theta)\eta\right]$$
$$\overset{(2)}{=} \theta_0 \sum_{j=1}^{J} \pi_j \mathbb{E}_{\beta^*, k, \eta}\left[\partial_y \hat{\beta}^{\mathrm{sc}}(\beta^* + \eta; k, \theta)\right], \tag{34}$$

where (1) arises from the fact that $\beta^*$ and $z$ are independent and $\mathbb{E}[z] = 0$ and (2) is a direct application of Stein's lemma,

$$\mathbb{E}[Xf(X)] = \sigma^2 \mathbb{E}[f'(X)] \text{ for } X \sim \mathcal{N}(0, \sigma^2). \tag{35}$$

Hence, we can reduce the original matrix-valued fixed point equations to a system of scalar fixed-point equations in the parameters $(p, \chi)$.

**Expressing $\partial_y \hat{\beta}(y; k, \theta)$.** Denoting

$$q_k'(x) := \frac{\partial q_k(x)}{\partial x}, \quad q_k''(x) := \frac{\partial^2 q_k(x)}{\partial^2 x^2}, \tag{36}$$

the first-order optimality condition for

$$\hat{\beta}^{\mathrm{sc}}(y; k, \theta) = \arg\min_{\beta} \frac{(y - \beta)^2}{2\theta} + q_k(\beta)$$

is

$$0 = \frac{1}{\theta}\left(\hat{\beta}^{\mathrm{sc}}(y; k, \theta) - y\right) + q_k'\left(\hat{\beta}^{\mathrm{sc}}(y; k, \theta)\right).$$

Implicit differentiation with respect to $y$ gives

$$\partial_y \hat{\beta}^{\mathrm{sc}}(y; k, \theta) = \frac{1}{1 + \theta\, q_k''\left(\hat{\beta}^{\mathrm{sc}}(y; k, \theta)\right)}.$$

We note that

$$q_k'(z) = \frac{1}{2}\operatorname{asinh}\left(\frac{2z}{\sqrt{k}}\right), \quad q_k''(z) = \frac{1}{\sqrt{k + 4z^2}}. \tag{37}$$

Substituting $\beta = \hat{\beta}^{\mathrm{sc}}(y; k, \theta)$ yields the explicit Jacobian

$$\partial_y \hat{\beta}^{\mathrm{sc}}(y; k, \theta) = \frac{1}{1 + \dfrac{\theta}{\sqrt{k + 4\hat{\beta}^{\mathrm{sc}}(y; k, \theta)^2}}}.$$

$$p = \sum_{j=1}^{J} \pi_j \mathbb{E}_{\beta^*, k, \eta}\left[\left(\hat{\beta}^{\mathrm{sc}}(\beta^* + \eta; k, \theta) - \beta^*\right)^2\right], \tag{38}$$

$$\chi = \theta_0 \sum_{j=1}^{J} \pi_j \mathbb{E}_{\beta^*, \eta}\left[\left(1 + \frac{\theta}{\sqrt{k + 4\hat{\beta}^{\mathrm{sc}}(\beta^* + \eta; k, \theta)^2}}\right)^{-1}\right], \tag{39}$$

with the closure relations

$$\theta = \frac{\lambda + \chi}{\alpha}, \quad \theta_0 = \frac{\sigma_0^2 + p}{\alpha}. \tag{40}$$

This proves the proposition.

# D. Experiments

In this section, we provide a more detailed description of the experiments conducted on ResNet architectures in different settings. We pair the accuracy results presented with a measure of the representation before and after fine-tuning. We employ the commonly used *participation ratio* (PR; (Gao et al., 2017)) as a measure of dimensionality, and the *effective number of shared dimensions* (ENSD; (Giaffar et al., 2024)) as a measure of the number of principal components aligned between two representations. Intuitively, the PR and ENSD of network representations before and after fine-tuning capture the key phenomenology of the pretraining-dependent rich regime. Specifically, we expect that the dimensionality of the network representation $X_{\mathrm{FT}}$ after fine-tuning is lower than that of the representation $X_{\mathrm{PT}}$ after pretraining, i.e., $\mathrm{PR}(X_{\mathrm{FT}}) < \mathrm{PR}(X_{\mathrm{PT}})$, and that nearly all representational dimensions expressed post-fine-tuning are inherited from the pretraining state, i.e., $\mathrm{ENSD}(X_{\mathrm{PT}}, X_{\mathrm{FT}}) \approx \mathrm{PR}(X_{\mathrm{FT}})$. The manifestation of this regime is more or less pronounced depending on the parameter we vary.

## D.1. Experiments $\lambda_{PT}$

The ResNet-18 architecture consists of an initial embedding layer followed by four stages of residual blocks, each containing two convolutional layers with identity skip connections. To induce imbalance across the network, we upscaled the embedding layer and the first three residual stages. In this experiment we modify lambda balanced during pretraining and leave the network unchanged during fine-tuning. In **Fig. 7** we observe that for decreasing $\kappa$ the representation in the last layer looks like a signature of the pretraining-dependent rich regime described above. Furthermore, we observe that the overall dimensionality of the network decreases as a function of $\kappa$.

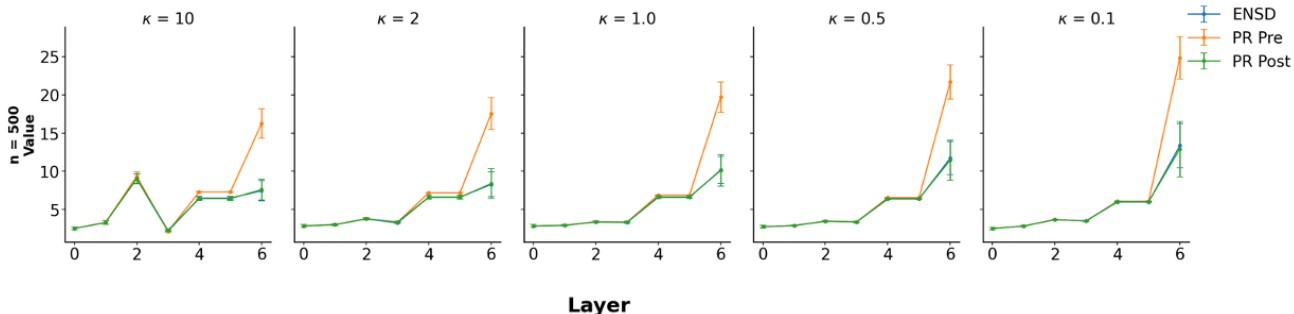

*Figure 7.* **ResNet experiments on CIFAR-100.** Resnet layers before and after fine-tuning (PR Pre and PR Post) as well as their ENSD as a function of the $\kappa_{FT}$ re-initialization.

## D.2. Experiments $c_{PT}$

To implement an equivalent notion of $c_{PT}$ scaling in this architecture, we scale all network parameters by a constant factor $\kappa$. In this experiment, the overall scaling is applied during pretraining, while the network remains unchanged during fine-tuning. In **Fig. 8** we observe that for decreasing $c_{PT}$ the representation in the last layer looks like a signature of the pretraining-dependent rich regime described above.

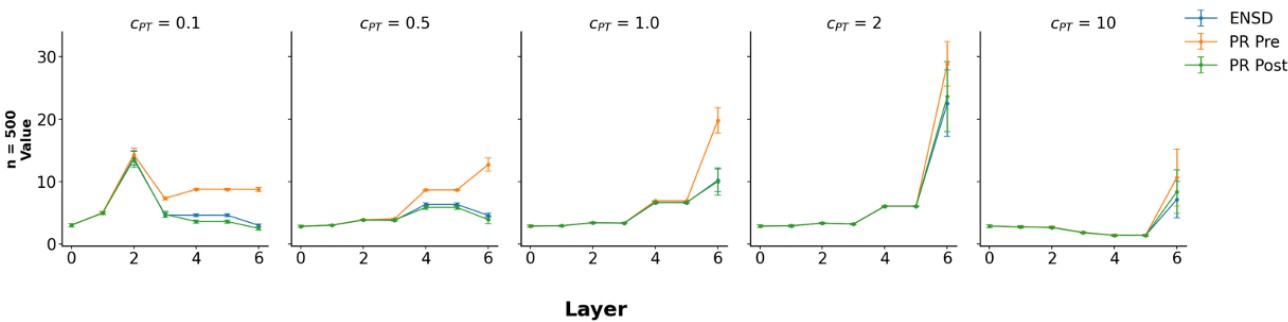

*Figure 8.* **ResNet experiments on CIFAR-100.** Resnet layers before and after fine-tuning (PR Pre and PR Post) as well as their ENSD as a function of the $c_{PT}$ re-initialization.

### D.3. Experiments $\gamma_{FT}$

To implement an equivalent notion of $\gamma_{FT}$ rescaling in this architecture, we scale the last layer parameter. In this experiment, the overall scaling is applied during fine-tuning, while the network remains unchanged during pretraining. In **Fig. 9** we observe that for decreasing $\gamma_{FT}$ the representation in the last layer looks like a signature of the pretraining-dependent rich regime discribed above.

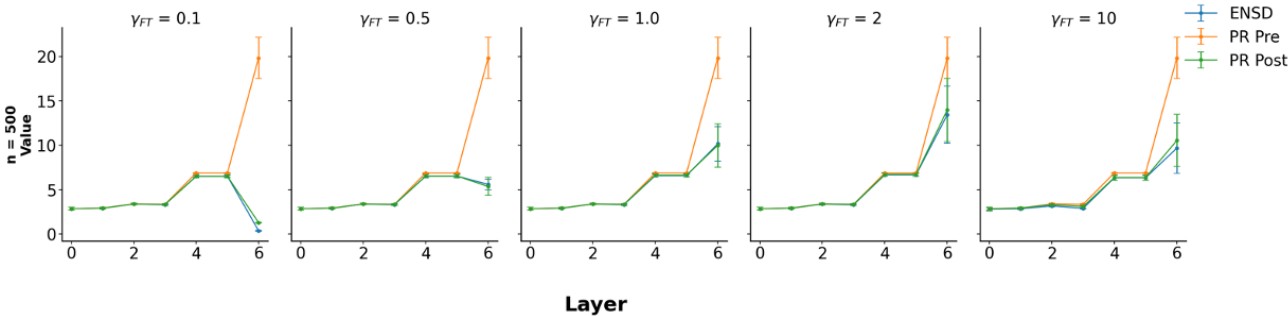

*Figure 9.* **ResNet experiments on CIFAR-100.** Resnet layers before and after fine-tuning (PR Pre and PR Post) as well as their ENSD as a function of the $\gamma_{FT}$ re-initialization.

### D.4. Experiments $c_{FT}$

For completeness, we include the heuristic proposed by (Lippl & Lindsey, 2024) for inducing a pretraining-dependent rich regime, which consists of rescaling all network weights by a constant $c_{FT} < 1$ during fine-tuning (see Appendix D). As shown in **Fig. 10**, this heuristic is reported to improve performance relative to the baseline. The values found are not the same as the one reported in (Lippl & Lindsey, 2024) since the set of parameter used are different. In **Fig. 10** we observe that for decreasing $c_{FT}$ the representation in the last layer looks like a signature of the pretraining-dependent rich regime described above.

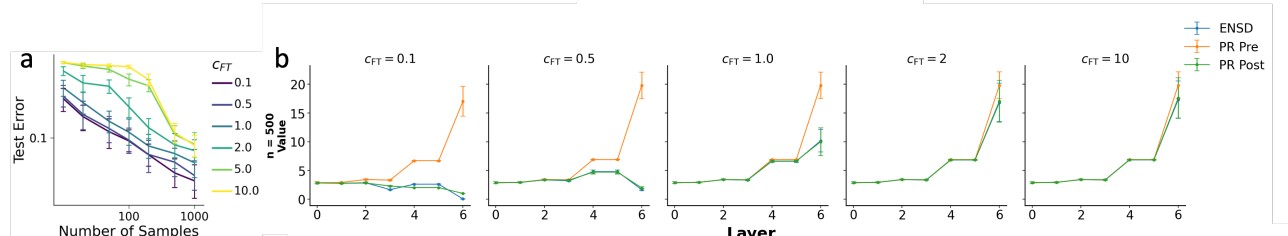

*Figure 10.* **ResNet experiments on CIFAR-100.** (a) Generalization performance as a function of the number of samples and initalization parameters. We vary $c_{FT}$. (b) Resnet layers before and after fine-tuning (PR Pre and PR Post) as well as their ENSD as a function of the $\gamma_{FT}$ re-initialization.

### D.5. Control of task similarity

While it is not straightforward to exactly measure task overlap, we investigate its impact by fine-tuning on a different dataset (SVHN), which is less similar to the PT data and therefore serves as a proxy for lower overlap (Fig. 11). FT performance exceeds that of single-task learning at intermediate training stages across all kappa values. Consistent with our CIFAR results, initializing ResNet with a smaller-than-standard scale enhances FT generalization. Overall, these findings support the hypothesis that reducing dependence on PT amplifies the advantages of scaling for tasks with lower overlap (compared to the CIFAR experiment in Fig.5 **a**).

### D.6. Method robustness to optimizer choice

We observe the same qualitative trend as diagonall networks in non-linear. In Fig. 11**b**, using a ResNet trained with Adam with weight decay, smaller $\alpha$ -our practical proxy for negative relative scale-consistently improves FT performance, indicating that the effect is not tied to a specific optimizer or model. While different optimizers introduce stochasticity, finite step sizes, and adaptive scaling, the qualitative dependence on relative scale and task structure remains unchanged,

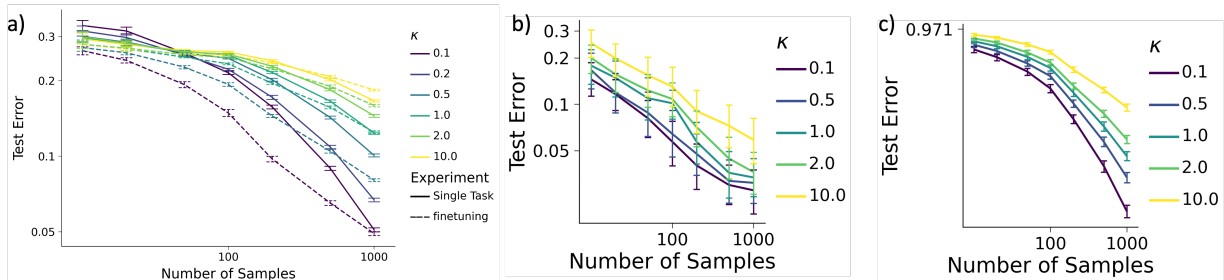

*Figure 11.* **Further ResNet experiments: (a)** *New SVHN Dataset*: After pretraining on 98 classes of CIFAR-100, we fine-tune the model on 2 classes of SVHN. Consistent with our CIFAR results, initializing ResNet with a non-standard small scale ($\kappa < 1$, denoting negative relative scaling) improves fine-tuning generalization. We contrast these results with the single task learning results on the two classes of SVHN and observe benefits of fine-tuning in the intermediate sample setting. We posit that the gap in performance gain (compared to CIFAR FT) might be smaller because the task overlap is smaller **(b)** *Adam Optimizer*: To test robustness across optimizers, we replace SGD with Adam (weight decay of $5 \times 10^{-3}$ during pre-training; decay towards initial weights during fine-tuning, inspired by our diagonal network findings). The $\kappa$ trend remains consistent, though slightly attenuated. **(c)** *Data Split*: We adjust the CIFAR-100 pretraining/fine-tuning split from 98/2 to 60/40. The benefits of a smaller $\kappa$ on generalization persist, confirming the robustness of the relative scaling effect under different task distributions.

indicating that the observed regimes are not optimizer-specific.

### D.7. Data split

Data Split: We vary the CIFAR-100 split from 98/2 (pretraining on 98 classes and fine-tuning on 2) to 60/40. Consistent with our CIFAR results, a smaller initialization scale () improves generalization (Fig. 5 **c**)

### D.8. Modular addition in Transformers

To assess whether our insights transfer to a different model architecture, we considered a novel PT+FT setup in Transformers. We pretrained a one-hidden-layer Transformer, expanding the codebase provided under the following link: `https://github.com/teddykoker/grokking`. The Transformer was pretrained on six different target outputs (using different linear readouts): for inputs $x, y$, these target outputs were $x \text{ op } y \mod b$, where $\text{op} \in \{\cdot, +\}$ and $b \in \{33, 145, 194\}$. We fine-tuned these models on modular addition using nine possible moduli: three seen moduli $(33, 145, 194)$, three novel moduli that share prime factors with the seen moduli $(55, 87, 97)$, and three moduli that don't share any prime factors $(49, 127, 169)$. We pretrained and fine-tuned all models using Adam with a learning rate of $4 \cdot 10^{-3}$, no weight decay, $\beta_1 = 0.9, \beta_2 = 0.98$. We pretrained for one million steps and fine-tuned for 32,000 steps, in order to investigate how quickly models are able to transfer their features to these different tasks. To examine the impact of balancedness, we multiplied the initial weights of the embedding by $\kappa$ (following Kunin et al. (2024)).

We found that smaller balancedness generally improves generalization performance during fine-tuning (Fig. 12a). Moreover, we also found that tasks involving only novel prime factors were noticeably harder (Fig. 12b).

## E. Implementation and Simulations

### E.1. Figure 2

In the code we approximate

$$\text{PD} = \frac{\partial \log P(k, \beta_{FT})}{\partial \log \beta_{PT}} = \frac{\partial \log P}{\partial \log k} \cdot \frac{\partial \log k}{\partial \log \beta_{PT}} = \frac{\partial logP}{\partial logk} \cdot \frac{\partial k}{\partial \beta_{PT}} \cdot \frac{\beta_{PT}}{k} \tag{41}$$

$$= \left[ \frac{1}{2} - \frac{\beta_{FT}}{\sqrt{k}} \cdot \frac{q'}{q} \right] \cdot \frac{\beta_{PT}}{k} \cdot \frac{\partial k}{\partial \beta_{PT}} \tag{42}$$

$$\tag{43}$$

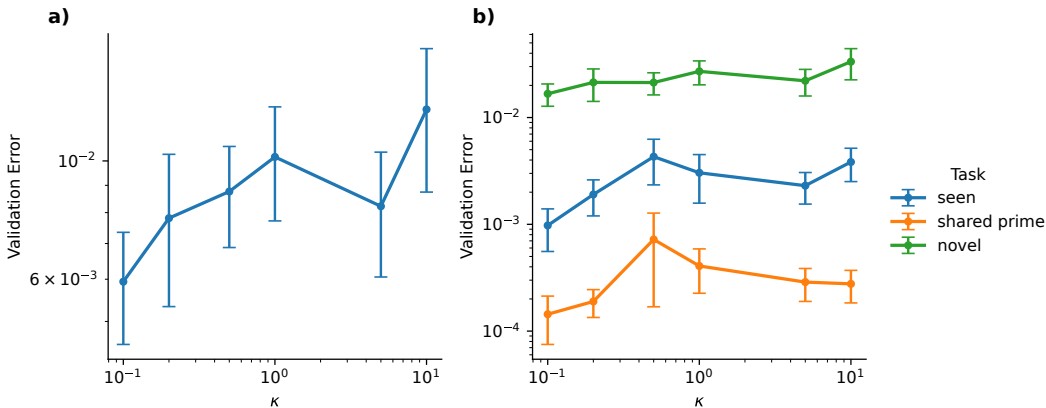

*Figure 12.* **Experiments on Transformers with modular multiplication.** We pretrain a one-hidden-layer Transformer on a mixture of modular multiplication and addition tasks using three moduli ($p = 33, 145, 194$). Following Kunin et al. (2024), we manipulate balancedness by multiplying the initial weights of the embedding by $\kappa$. We then fine-tune the Transformer on modular multiplication with nine distinct moduli, to test varying degrees of task overlap. Specifically, we consider the seen moduli ($p = 33, 145, 194$), in which case the model must select the relevant subset of pretraining features; moduli that share prime factors ($p = 55, 87, 97$), in which case the model can similarly rely on a set of overlapping features; and moduli that do not share any prime factors ($p = 49, 127, 169$), in which case the model may be able to transfer some features but may also have to learn a set of novel features. We ran 25 random seeds per setup; error bars reflect $\pm$ one standard error. **a)** Best validation error as a function of $\kappa$. Smaller $\kappa$ generally performs better. **b)** Best validation error averages within the different categories of fine-tuning tasks. Tasks involving only novel prime factors are noticeably harder.

with

$$\frac{\partial k}{\partial \beta_{PT}} \approx \frac{k_+ - k_-}{2\epsilon}. \tag{44}$$

$$k_+ = 2(\lambda_{\text{PT}} + c_{\text{PT}})\left(1 + \sqrt{1 + ((\beta_{PT} + \epsilon)/c_{\text{PT}})^2}\right) + \gamma_{FT}^2 \tag{45}$$

$$k_- = 2(\lambda_{\text{PT}} + c_{\text{PT}})\left(1 + \sqrt{1 + ((\beta_{PT} - \epsilon)/c_{\text{PT}})^2}\right) + \gamma_{FT}^2 \tag{46}$$

### E.2. Figure 3

#### E.2.1. SOLVING THE REPLICA FIXED POINT EQUATIONS NUMERICALLY

The RS order parameters are

$$p = \mathbb{E}\left[(\beta_{\text{ft}} - \hat{\beta}_{\text{ft}})^2\right], \qquad \chi = \theta_0 \mathbb{E}\left[\partial_y \hat{\beta}^{\text{sc}}(y; K, \theta)\right],$$

where the expectation is over $(\beta_{\text{ft}}, K, v)$ with $y = \beta_{\text{ft}} + \sigma v$, $v \sim \mathcal{N}(0,1)$, and $\sigma^2 = \theta_0 = (p + \sigma_0^2)/\alpha$.

The fixed-point closure is

$$\theta_0 = \delta(\sigma_0^2 + p), \qquad \theta = \delta(\chi + \lambda),$$

with $\delta = 1/\alpha$, additive label-noise variance $\sigma_0^2$ (set to zero in the noiseless teacher setting), and an optional external ridge parameter $\lambda$.

These equations are solved iteratively for $(\theta_0, \theta)$ at each $\alpha$. In order to solve each equation, we need to evaluate to separate expectations.

The expectations defining $p$ and $\chi$ are approximated by Monte-Carlo sampling. Given $m$ i.i.d. samples $(\beta_{\text{ft},i}, k_i, v_i)$, we form scalar observations

$$y_i = \beta_{\text{ft},i} + \sqrt{\theta_0}\, v_i, \qquad v_i \sim \mathcal{N}(0,1).$$

For each $y_i$ and penalty parameter $k_i$, the scalar estimator $\hat{\beta}_{\text{ft},i}$ is given by the RS scalar denoiser

$$\hat{\beta}_{\text{ft},i} = \hat{\beta}^{\text{sc}}(y_i; k_i, \theta) = \arg\min_{\beta \in \mathbb{R}} \left\{ \frac{(y_i - \beta)^2}{2\theta} + q_{k_i}(\beta) \right\}.$$

This scalar optimization balances fidelity to the noisy observation $y_i$ against the implicit-bias penalty $q_{k_i}$, with $\theta$ controlling the strength of the quadratic term. For the $q_k$ family considered here, the objective is strictly convex, so the minimizer is unique and can be computed reliably via a safeguarded Newton method.

At the scalar optimum, we also compute the local curvature

$$s_i^2 := \left(\tfrac{1}{\theta} + q_{k_i}''(\hat{\beta}_{\text{ft},i})\right)^{-1},$$

where $\hat{\beta}_{\text{ft},i} = \hat{\beta}^{\text{sc}}(y_i; k_i, \theta)$ and $y_i = \beta_{\text{ft},i} + \sqrt{\theta_0}\, v_i$. This expression follows by implicit differentiation of the first-order optimality condition for the scalar denoiser:

$$0 = \frac{\hat{\beta}^{\text{sc}}(y; k, \theta) - y}{\theta} + q_k'\big(\hat{\beta}^{\text{sc}}(y; k, \theta)\big).$$

Differentiating both sides with respect to $y$ gives

$$0 = \frac{1}{\theta}\Big(\partial_y \hat{\beta}^{\text{sc}}(y; k, \theta) - 1\Big) + q_k''\big(\hat{\beta}^{\text{sc}}(y; k, \theta)\big)\, \partial_y \hat{\beta}^{\text{sc}}(y; k, \theta),$$

and therefore

$$\partial_y \hat{\beta}^{\text{sc}}(y; k, \theta) = \frac{1}{1 + \theta\, q_k''(\hat{\beta}^{\text{sc}}(y; k, \theta))} = \frac{s^2}{\theta}, \qquad s^2 = \left(\tfrac{1}{\theta} + q_k''(\hat{\beta}^{\text{sc}})\right)^{-1}.$$

We compute $s_i^2$ (equivalently $\partial_y \hat{\beta}^{\text{sc}}$) at the optimum because it provides a numerically stable evaluation of the susceptibility: it only involves the positive quantity $\tfrac{1}{\theta} + q_{k_i}''(\hat{\beta}_{\text{ft},i})$ (which is bounded away from zero under strict convexity), avoiding finite-difference approximations of $\partial_y \hat{\beta}^{\text{sc}}$ that can be noisy or ill-conditioned.

The Monte-Carlo estimates of the RS moments are then

$$\hat{p} = \frac{1}{m}\sum_{i=1}^{m}(\beta_{\text{ft},i} - \hat{\beta}_{\text{ft},i})^2, \qquad \hat{\chi} = \theta_0 \cdot \frac{1}{m}\sum_{i=1}^{m} \partial_y \hat{\beta}^{\text{sc}}(y_i; k_i, \theta) = \theta_0 \cdot \frac{1}{m}\sum_{i=1}^{m}\frac{s_i^2}{\theta},$$

which are substituted into the fixed-point closure relations and iterated (with damping) until convergence.

When the penalty parameter $k$ takes values in a finite set, the Monte-Carlo evaluation can be accelerated by grouping samples with identical $k$. In the PT→FT oracle considered here, $k$ is deterministic conditional on PT activity: by construction (and because $\alpha_{\text{pt}} \geq 1$ in our setting), pretraining recovers the teacher exactly so each coordinate has $\hat{\beta}_{\text{pt},d} \in \{0, \pm 1/\sqrt{\rho_{\text{pt}}}\}$. Since $k_d$ is a deterministic function of $\hat{\beta}_{\text{pt},d}$ and the initialization hyperparameters (cf. Eq. (15)), it follows that $k_d$ can only take finitely many values (one for PT-inactive coordinates and one for PT-active coordinates, or more generally one per PT group if multiple groups are used). Importantly, while the proximal solution $\hat{\beta}^{\text{sc}}(y_i; k_i, \theta)$ depends on the sampled $\beta_{\text{ft},i}$ through $y_i$, the parameter $k_i$ itself does not: it is fixed by the pretrained coordinate type (PT-active vs. PT-inactive). Therefore grouping by identical $k$ is valid even though $\hat{\beta}_{\text{ft},i}$ varies across samples within a group.

### E.2.2. Fixed-point iteration and numerical stabilization

Although the replica-symmetric fixed-point equations are theoretically well defined, naive numerical iteration can be unstable, especially near sharp transitions or in regimes with multiple admissible solutions. We therefore employ a damped fixed-point scheme with simple numerical safeguards to ensure stable and reproducible convergence.

**Iteration variables.** For a fixed inverse sample efficiency $\delta = 1/\alpha$, the solver iterates on the RS state variables $(\theta_0, g)$. Given a current iterate, Monte-Carlo estimates of the RS moments $\hat{p}$ and $\hat{\chi}$ are computed as described in Section 2 and used to form the updates

$$\theta_0^{\text{new}} = \sigma_0^2 + \delta\hat{p}, \qquad g^{\text{new}} = \gamma_{\text{ext}} + \delta g \hat{\chi}.$$

A fixed point of this map defines the RS solution at the given $\alpha$.

**Damping.** To suppress oscillations and divergence, the updates are applied with damping parameter $\lambda \in (0, 1]$:

$$\theta_0 \leftarrow (1 - \lambda)\theta_0 + \lambda\theta_0^{\text{new}}, \qquad g \leftarrow (1 - \lambda)g + \lambda g^{\text{new}}.$$

Smaller values of $\lambda$ slow convergence but substantially improve stability, particularly near phase transitions.

**Positivity constraints.** The RS variables are constrained to remain in their admissible domains by enforcing

$$\theta_0 \geq \sigma_0^2, \qquad g \geq g_{\min},$$

where $g_{\min} > 0$ is a small numerical floor. These constraints prevent degeneracy in the scalar denoiser and curvature evaluation and act purely as numerical safeguards. I practice we pick $g_{\min} = 10^{-14}$

**Convergence criteria.** Convergence is assessed using the maximum absolute residual

$$\text{res} = \max\{|\theta_0^{\text{new}} - \theta_0|, \ |g^{\text{new}} - g|\}.$$

The iteration terminates when $\text{res} < \texttt{tol}$ or when a preset iteration limit is reached.

### E.2.3. STABILIZATION ACROSS SAMPLE EFFICIENCIES

In addition to within-$\alpha$ stabilization, the RS fixed-point equations may admit multiple stable solutions as the sample efficiency $\alpha$ varies. To robustly track solutions across $\alpha$, we use continuation with warm starts and a simple branch-selection rule.

**Forward/backward continuation in $\alpha$.** Given a grid of sample efficiencies $\{\alpha_j\}$ (equivalently $\delta_j = 1/\alpha_j$), we solve the fixed-point equations sequentially in two passes. In the forward pass, solutions at $\alpha_j$ are initialized using the converged state from $\alpha_{j-1}$; in the backward pass, the grid is traversed in reverse order, initializing from $\alpha_{j+1}$. This bidirectional continuation helps detect multistability and reduces sensitivity to initialization.

We explicitly verify that the forward and backward continuations converge to consistent solutions across the entire grid. For each $\alpha_j$, we compare the converged forward and backward fixed points and observe no evidence of multistability. Quantitatively, the median branch mismatch $\|\theta_{\text{fwd}} - \theta_{\text{bwd}}\|$ is $\sim 1.5 \times 10^{-7}$ (with the 95th percentile below $2 \times 10^{-6}$), and the corresponding fixed-point residuals are of order $10^{-7}$–$10^{-6}$. The resulting Monte-Carlo estimates of $p$ and $\chi$ differ by less than $1.5\%$ in median and $2.5\%$ at the 95th percentile, well within the intrinsic sampling error.

In addition, the implementation includes an automated reliability score that flags numerical instabilities; no runs failed to converge or exhibited exploding behavior across all sweeps. A small MSE floor ($\sim 10^{-12}$) is used to prevent numerical issues in the high-sample-efficiency regime, where such instabilities are known to arise. Taken together, these checks confirm that the fixed-point solutions are robust, path-independent, and insensitive to the direction of continuation in $\alpha$.

**Warm starts.** At each $\alpha_j$, the fixed-point iteration is warm-started from the nearest converged solution along the continuation path, rather than from a generic initialization. This reduces the number of iterations required for convergence.

**Branch selection rule.** Forward and backward continuation may converge to different fixed points at the same $\alpha$, reflecting genuine RS multistability. In such cases, we select the branch with smaller predicted MSE as the reported solution. The discrepancy between forward and backward solutions is retained as a diagnostic of numerical sensitivity.

### E.2.4. DIAGNOSTICS AND RELIABILITY CHECKS

To assess the reliability of numerical RS solutions, we record diagnostics that quantify fixed-point convergence, multistability across continuation paths, and Monte-Carlo uncertainty.

**Residual checks.** For each $\alpha$, convergence is monitored using the fixed-point residual

$$\text{res} = \max\{|\theta_0^{\text{new}} - \theta_0|, \ |g^{\text{new}} - g|\}.$$

Solutions with $\text{res} > \texttt{tol}$ are considered unconverged and flagged as unreliable.

**Branch mismatch.** Let $\text{MSE}_{\text{fwd}}(\alpha)$ and $\text{MSE}_{\text{bwd}}(\alpha)$ denote the RS predictions obtained from forward and backward continuation, respectively. We quantify branch disagreement via

$$\Delta_{\text{branch}}(\alpha) = \left|10 \log_{10}\!\big(\text{MSE}_{\text{fwd}}(\alpha)\big) - 10 \log_{10}\!\big(\text{MSE}_{\text{bwd}}(\alpha)\big)\right|.$$

Large values indicate RS multistability or sensitivity to initialization.

**Monte-Carlo uncertainty.** Let

$$e_i^2 = \left( \beta_{\mathrm{ft},i} - \hat{\beta}_{\mathrm{ft},i} \right)^2.$$

The Monte-Carlo estimate

$$\widehat{\mathrm{MSE}} = \frac{1}{m} \sum_{i=1}^{m} e_i^2$$

is assigned a standard error $\mathrm{SE}(\widehat{\mathrm{MSE}})$, computed using batch-means estimation. Uncertainty on the log scale is approximated via the delta method,

$$\mathrm{SE}_{\mathrm{dB}} \approx \frac{10}{\ln 10} \frac{\mathrm{SE}(\widehat{\mathrm{MSE}})}{\max\{\widehat{\mathrm{MSE}}, \varepsilon\}},$$

with a small floor $\varepsilon > 0$ to avoid numerical blow-up.

**Failure indicators.** A solution at $\alpha$ is flagged as unreliable if any of the following occur:

- res $>$ tol: the fixed-point residual exceeds the prescribed tolerance. In all experiments, the residual remained strictly below tol $= 10^{-10}$, with a maximal observed value of approximately $9.9 \times 10^{-7}$.

- $\Delta_{\mathrm{branch}}(\alpha)$ exceeds a prescribed threshold: the mismatch between forward and backward continuation branches becomes large. Across all sweeps, the maximal observed branch mismatch ($\Delta_{\mathrm{branch}}$, measured in dB) was approximately $3.6 \times 10^{-4}$ dB, occurring in the most challenging low-$\alpha$ regimes, and is negligible relative to the reported MSE effects.

- $\mathrm{SE}_{\mathrm{dB}}$ is comparable to or larger than the reported effect size: the Monte Carlo standard error of the predicted MSE (in dB) becomes large. In practice, $\mathrm{SE}_{\mathrm{dB}}$ was capped by a numerical floor and remained below approximately $0.07$ dB even in the low-$\alpha$ regime, ensuring that uncertainty in the RS predictions remains well controlled as $\mathrm{MSE} \to 0$.

These indicators are retained alongside RS predictions to identify unstable or poorly resolved regions. In the current experimental setups, no solutions exceeded these reliability thresholds.

The consistently small values of all failure indicators can be attributed to (i) strict convergence criteria combined with conservative damping and a large iteration budget, (ii) the absence of detectable multistability in the explored parameter regimes, as evidenced by near-identical forward and backward continuations, and (iii) explicit slope capping in the dB-scale error estimation, which prevents numerical amplification of Monte Carlo uncertainty at very small MSE.

### E.2.5. PARAMETER SWEEPS AND CURVE GENERATION

All replica-theory curves are generated by solving the RS fixed-point equations over a dense grid of sample efficiencies $\alpha$, mirroring the structure of the corresponding empirical experiments.

**Numerical parameters.** Replica fixed-point equations are solved using Monte-Carlo approximation with $m = 80{,}000$ samples per $\alpha$. Fixed-point iteration uses damping factor $0.25$, convergence tolerance tol $= 10^{-6}$, and a maximum of $900$ iterations per $\alpha$. A small external ridge parameter $\gamma_{\mathrm{ext}} = 10^{-6}$ is included to improve numerical stability. All runs use a single random seed.

**Parameter sweeps.** For each experiment, a baseline configuration is evaluated together with one-dimensional sweeps over individual hyperparameters, with all remaining parameters held fixed. Baseline values are excluded from sweep lists to avoid duplicate runs. Each parameter configuration produces a full generalization curve indexed by $\alpha$.

**Parallelization and curve assembly.** To increase parallelism, the $\alpha$-grid may be partitioned into contiguous chunks, with each job solving the RS equations on a subset of $\alpha$ values for a fixed parameter configuration. Results from all chunks are concatenated to form the final curve, which is saved together with per-$\alpha$ diagnostics and metadata.

### E.2.6. DIAGONAL NETWORK EXPERIMENTS (EMPIRICAL SETUP)

This section specifies the empirical diagonal-network experiments used to compare finite-dimensional training with the replica-symmetric predictions. All experiments use diagonal linear networks trained by gradient flow and are averaged over multiple random seeds.

**Global settings.** Unless stated otherwise, all experiments use the following hyperparameters:

| Parameter | Value |
|---|---|
| Input dimension $d$ | 5000 |
| Test samples | 10,000 |
| Learning rate | 0.5 |
| Max epochs | $5 \times 10^6$ |
| Convergence threshold | $10^{-4}$ |
| Data scale $\alpha = n/d$ | 11 values in $[0.01, 0.5]$ |
| Random seeds | 14 (seeds 6–19) |
| Teacher signal scale $a_{PT}$ | 1.0 |
| Pretraining sparsity $\rho_{PT}$ | 0.1 |

Mean squared error is evaluated on an independent test set after convergence.

**Task parameterization and overlap.** Fine-tuning task structure is specified by $(\rho_{FT}^{\text{shared}}, \rho_{FT}^{\text{new}})$ as in Eq. (11), with total fine-tuning sparsity

$$\rho_{FT} := \rho_{FT}^{\text{shared}} + \rho_{FT}^{\text{new}}.$$

For convenience, we additionally report the *overlap fraction*

$$\omega := \frac{\rho_{FT}^{\text{shared}}}{\rho_{FT}^{\text{shared}} + \rho_{FT}^{\text{new}}} \quad \in [0, 1],$$

so that, for a fixed $\rho_{FT}$, sweeping $\omega$ corresponds to setting $\rho_{FT}^{\text{shared}} = \omega \, \rho_{FT}$ and $\rho_{FT}^{\text{new}} = (1 - \omega) \, \rho_{FT}$.

**Pretrain–fine-tune protocol.** For pretrain–fine-tune (PT+FT) experiments, fine-tuning is initialized from an analytically constructed infinite-pretraining state determined by the pretraining teacher $\beta_{PT}$ and homogeneous parameters $(c_{PT}, \lambda_{PT})$. Here we assume $\alpha_{PT} \geq 1$ so that pretraining perfectly recovers the ground truth $\beta_{PT}$. At the start of fine-tuning, a reinitialization scale $\gamma_{\text{reinit}}$ is applied to set $\beta(0) \approx 0$ while preserving the coordinate-wise richness structure.

**Experiment 1: Benefit from existing features.** Fixed parameters:

$$\rho_{PT} = 0.1, \qquad \rho_{FT} = \rho_{FT}^{\text{shared}} + \rho_{FT}^{\text{new}} = 0.1.$$

Baseline:

$$\omega = 0.5 \ \ (\text{i.e., } \rho_{FT}^{\text{shared}} = \rho_{FT}^{\text{new}} = 0.05), \quad c_{PT} = 10^{-3}, \quad \lambda_{PT} = 0, \quad \gamma_{\text{reinit}} = 0.$$

Swept parameters:

| Parameter | Values |
|---|---|
| Overlap fraction $\omega$ | $\{0, 0.5, 1\}$ |
| $c_{PT}$ | $\{10^{-6}, 10^{-3}, 1\}$ |
| $\lambda_{PT}$ | $\{-10^{-3}, -0.99 \cdot 10^{-3}, 0, 0.99 \cdot 10^{-3}\}$ |
| $\gamma_{\text{reinit}}$ | $\{0, 1, 10\}$ |

**Experiment 2: Learning new features.** Fixed parameters:

$$\rho_{PT} = 0.1, \qquad \rho_{FT}^{\text{shared}} = 0 \quad (\text{equivalently } \omega = 0).$$

Swept parameters:

| Parameter | Values |
|---|---|
| $\rho_{FT}^{\text{new}}$ | $\{0.1, 0.9\}$ |
| ($\rho_{FT} = \rho_{FT}^{\text{new}}$ since $\rho_{FT}^{\text{shared}} = 0$) | |
| $c_{PT}, \lambda_{PT}, \gamma_{\text{reinit}}$ | same as Experiment 1 |

**Experiment 3: Nested feature regime.**   Fixed parameters:

$$\rho_{PT} = 0.1.$$

Swept parameters:

| Parameter | Values |
|---|---|
| Overlap fraction $\omega$ | $\{0, 1\}$ |
| $\rho_{FT}$ | $\{0.01, 0.04\}$ |
| Implied $(\rho_{FT}^{\text{shared}}, \rho_{FT}^{\text{new}})$ | $\omega = 1: \ (\rho_{FT}^{\text{shared}}, \rho_{FT}^{\text{new}}) = (\rho_{FT}, 0)$ 
 $\omega = 0: \ (\rho_{FT}^{\text{shared}}, \rho_{FT}^{\text{new}}) = (0, \rho_{FT})$ |
| $c_{PT}, \lambda_{PT}, \gamma_{\text{reinit}}$ | same as Experiment 1 |

**Experiment 4: Single-task learning (SLT).**   In this setting, the network is trained from scratch without pretraining. We generate a single sparse task with sparsity $\rho$ (denoted $\rho_{PT}$ for notational consistency).

Swept parameters:

| Parameter | Values |
|---|---|
| Task sparsity $\rho$ (denoted $\rho_{PT}$) | $\{0.01, 0.04, 0.1, 0.9\}$ |
| $c_{PT}$ | $\{10^{-6}, 10^{-3}, 1\}$ |
| $\lambda_{PT}$ | $\{0, -c_{PT}, -0.99c_{PT}, 0.99c_{PT}\}$ |

**Mapping to replica curves.**   Empirical results are compared to replica-symmetric predictions by mapping the initialization at the start of fine-tuning to the replica penalty parameter

$$k_i = 4 \, c_{FT,i}^2,$$

and solving the associated fixed-point equations for each data scale $\alpha$.

**Code.**   The experiments and replica curves are generated using:

- `ptft_empirical_finetune_df.py`

- `ptft_replica_qk.py`

- `compute_emp_curves_worker_exp[1--4].py`

- `ExperimentSetup.md`

### E.3. Figure 5

All experiments are performed using ResNet-18. Each plot is based on 30 random seeds, with the mean performance shown and standard errors represented as error bars. The network is pre-trained on 49,000 samples until the loss reaches 0.01. During fine-tuning, the number of samples is varied as indicated in the figure, using a loss threshold of 0.0001. For the corresponding ENSD experiments, we use 500 samples.

