# OpenReview forum: "A Theory of How Pretraining Shapes Inductive Bias in Fine-Tuning"
_ICML.cc/2026/Conference — ICML 2026 regular_

### Official Review · Reviewer_PQKQ · 2026-03-09

**Soundness:** 3
**Presentation:** 2
**Significance:** 3
**Originality:** 3
**Overall Recommendation:** 5
**Confidence:** 3

**Summary:**

This work develops a theoretical framework for the commonly used pretraining-finetuning pipeline by

1. deriving the implicit bias of the final finetuned function learned for diagonal linear networks, and
2. identifying and analyzing feature learning regimes based on specific quantities related to parameter initialization – primarily the absolute and relative weight scales, and the re-initialization scheme of “readout” weights.

Finally, theoretical predictions are empirically validated on a simple ResNet18+CIFAR-100 setup.

**Compliance With Llm Reviewing Policy:**

Affirmed.

**Final Justification:**

I have updated my score from 4 to 5 in light of the additional experiments, justifications, and actions that the authors plan to add in the revised version.

**Key Questions For Authors:**

- In Fig 4(a), the blue line is lower than the violet line, which breaks the general trend that is seen across figures. I assume this is due to some tension between weight initialization schemes and the initialization parameters studied in this work -- can you comment on the relationship between them?

- In the same vein, how can one think about applying the results of this work to practical settings -- to me, this work seems to hold potential in that respect! For instance, how should we be thinking about weight (re-)initialization in light of the findings of this work?

- See question on task overlap in Weaknesses.

**Limitations:**

Yes.

**Strengths And Weaknesses:**

*Strengths*

- The research questions that the paper addresses are interesting and very relevant. In particular, I appreciate seeing a theoretical analysis that goes beyond a single stage of training, even if in a simplified setting that does not exactly match what is done in practice.

- The main findings of the paper are quite clear and it does a good job of providing cues to support intuition where needed.

- While most experiments are largely toy problems, they are conducted rigorously over multiple trials to illustrate that the theoretical results indeed hold.

- The paper does a great job of discussing the limitations of this work (albeit mostly in the Appendix).

*Weaknesses*

- The written structure and in general, the clarity of the paper could be improved. For instance, Fig. 1 did not particularly provide a better understanding of how diagonal linear networks work, and the main results of Sections 4 and 5 seem disconnected from each other due to the absence of proofs -- the latter is understandable due to space constraints. A few more high-level details on how Section 4 flows into Section 5 would help.

- A more detailed discussion on future directions, motivations, and why they would be worth pursuing would strengthen the paper.

- The experiment with ResNet18+CIFAR-100, while valuable, remains a bit unconvincing due to (a) the pretraining and finetuning task being pulled from the same dataset (albeit the classes are mutually exclusive) and (b) the massive difference in the sizes of the pretraining and finetuning datasets. An additional experiment that uses different datasets to pretrain and finetune to show if and how results deviate would make for a more convincing set of results.

- The notion of task overlap as defined makes sense, but also seems limited – could one consider something akin to a functional similarity or distribution shift to model this instead of dimensions?

---

> ### Author Rebuttal · Authors · 2026-03-31
>
> We thank the reviewer for their thoughtful comments. We link to additional experiments (Figures) discussed in the rebuttal [here](https://shorturl.at/c11PL).
>
> > **Clarity of the paper could be improved**
>
> Thank you for the suggestions. We will expand Fig. 1 to explain our setup in more detail. Specifically, one panel will describe the PT+FT setup in diagonal linear networks, focusing only on describing the architecture and the different initialization parameters. In a second panel, we will then describe our teacher-student setup, illustrating the task parameters (i.e. PT and FT sparsity as well as their overlap) and the training procedure. We hope that this will make the schematic in the figure clearer and are happy to address any further suggestions.
>
> Additionally, to improve the flow between Sections 4 and 5, we will outline the expected impact on generalization error and provide clearer predictions on the effects of overlap and sparsity based on the implicit bias in Section 4. This should better connect to Section 5, where we further emphasize the transition.
>
> > **Different dataset to finetune and different finetuning pretraining**
>
> Following the reviewer’s suggestion, we study FT on a new task (Fig. 5a, linked pdf). After PT on 98 CIFAR-100 classes, we fine-tune on 2 SVHN classes. Consistent with our CIFAR results, using a smaller initialization scale ($\kappa < 1$) improves generalization. We also vary the CIFAR-100 split from 98/2 to 60/40. The generalization gains from smaller $\kappa$ persist, confirming the robustness of the relative scaling effect  (Fig. 5c, linked pdf). For further details, please see the relevant part of the answer to reviewer r9Q8.
>
> > **Expanding the notion of task overlap**
>
> Yes, we think such extensions would be very interesting. One example of a concrete measure would be to consider the features used to implement each task in isolation (e.g. by training a network on a limited amount of data for each task) and then compute the effective number of shared dimensions, which is essentially a general measure of feature similarity. Notably, in the case of diagonal linear networks, this would directly amount to our current measure of task overlap, thus providing a natural extension.
>
> Beyond such formalized measures, we also expect that more task-specific intuitive measures could play a similar role. To illustrate this, we highlight a novel experiment we ran in response to a different reviewer's suggestion, studying pretraining and FT on modular multiplication tasks (Fig 6, linked pdf). In this case, we observed that FT on new moduli with shared prime factors (intuitively inducing a higher degree of overlap) yielded much better performance than FT on new moduli that did not share any prime factors.
>
> > **A more detailed discussion on future directions, motivations, and why they would be worth pursuing would strengthen the paper.**
>
> Great point. Our paper opens up several important avenues for investigation. First, while we focused on a scenario with one pretraining and one FT task, a similar analysis could be conducted in a continual learning paradigm. Such an analysis would allow us to better understand the conditions on both task similarity and initialization that allow for good forward and backward transfer. Second, it would be interesting to extend the analysis provided here to nonlinear neural networks, as nonlinear tasks can, in particular, exhibit substantially more complicated kinds of feature overlap. Finally, future work could leverage the implicit inductive biases highlighted in this work in order to design explicit regularization objectives that drive the network into the different identified regimes. We will add this discussion to the paper.
>
> > **In Fig 4(a), the blue line is lower than the violet line**
>
> Thank you for pointing this out. First, we note that the performance reversal between ($c_{PT} = 0.1$) and ($c_{PT} = 0.5$) is not statistically significant. It may nevertheless reflect a trade-off between initialization scale and optimizer dynamics. Since overlap and sparsity are not controlled for in this setting, they may also contribute—particularly if the task benefits from the network retaining some more general features rather than adjusting its features fully to the pretraining task (i.e. some laziness). We will add this discussion to the paper.
>
> > **How can one think about applying the results of this work to practical settings?**
>
> Thank you for this question. We agree that the potential practical implications of our insights are quite intriguing; we refer you to the relevant part of our response to Reviewer r9Q8 who raised a similar question. Overall, we thank you for your useful questions and in particular your helpful suggestions on improving our set of practical experiments and our presentation. We will implement them in the revised version of the manuscript and hope that they have addressed your concerns.

---

> > ### Author Rebuttal · Reviewer_PQKQ · 2026-04-01
> >
> > I thank the authors for their well-considered responses.

---

### Official Review · Reviewer_r9Q8 · 2026-03-09

**Soundness:** 3
**Presentation:** 3
**Significance:** 3
**Originality:** 3
**Overall Recommendation:** 4
**Confidence:** 4

**Summary:**

This paper develops an analytical theory of how pretraining initialization shapes inductive bias during fine-tuning. The core contribution is identifying four fine-tuning regimes induced by initialization scales, the theory formalizes the trade-off between feature reuse and feature learning, and empirical results suggest theoretical trends transfer to nonlinear networks.

**Compliance With Llm Reviewing Policy:**

Affirmed.

**Final Justification:**

The authors give a reasonable clarification, I prefer to keep my positive ratings and raise the confidence.

Such a theoretical contribution is valuable since "PT-FT" has become the mainstream paradigm, but I think it is still necessary to validate its effectiveness on more common and larger architectures, such as ViT, DINO, and CLIP, even the modern MLLMs.

**Key Questions For Authors:**

1. What are the applicability limits in deep nonlinear architectures?

2. Can adaptive initialization strategies be designed from the theory?

**Limitations:**

1. Lack of large-scale model validation. As this study lies in the background of pretraining-finetuning paradigms, the larger datasets and model size settings (\eg, ViT) are also required..

**Strengths And Weaknesses:**

Strengths:

1. End-to-end theoretical pipeline. This work connects implicit bias derivation, generalization analysis, and experiments.

2. Novel emphasis on relative initialization scale.

Weakness:

1. Simplified theoretical model. Diagonal linear networks ignore nonlinearities, feature interactions, and deep structure.

2. Simplified task assumptions. Spike-and-slab sparse generative model may not reflect real feature distributions.

---

> ### Author Rebuttal · Authors · 2026-03-31
>
> We thank the reviewer for their thoughtful comments. We link to additional experiments (Figures) discussed in the rebuttal [here](https://shorturl.at/c11PL).
>
> > **Simplified model**
>
> Please refer to our response to Reviewer EU24 (‘significance’ paragraph) where we further justify our choice of a linear network model.
>
> > **What are the applicability limits in deep nonlinear architectures?**
>
> While our main paper we focused on ResNet PT+FT on CIFAR-100 and its subsets, in this rebuttal we extend our analysis to a different dataset, alternative optimizers, new architectures, and new task splits:
>
> 1. **Different dataset (SVHN):** We study the effect of PT on CIFAR-100 and FT on SVHN (Fig. 5a in the attached document). We suggest that this operationalizes a lower degree of task overlap, as we fine-tune on a different dataset rather than different classes from the same dataset. We find that fine-tuning yields improved performance over single-task learning and that smaller relative scale in earlier layers ($\kappa<1$) improves the benefits of fine-tuning. Notably, the benefits of $\kappa<1$ are even stronger in this case compared to the CIFAR-to-CIFAR transfer experiment. This is consistent with our theory, as the lower degree of task overlap means that the network should benefit further from the fact that $\kappa<1$ makes the regime less pretraining-dependent and makes feature learning easier.
> 2. **Alternative optimizers (Adam):** In Fig. 5b, using a ResNet trained with Adam (L2P regularisation on FT), smaller $\kappa$ (our practical proxy for negative relative scale) consistently improves FT performance, indicating that the effect is not tied to a specific optimizer or model.
> 3. **Alternative architectures (Transformers):** To expand the range of our model architectures, we additionally ran experiments on small Transformers learning a modular multiplication task (adapting the setup in Kunin et al. (2024)). Specifically, we considered a broad PT task: modular multiplication and addition for different moduli (p=33, 145, 197; each of the six outputs was produced by a different readout dimension). We then fine-tuned on modular multiplication involving different moduli (the seen moduli: p=33, 145, 197; moduli with shared prime factors: p=55, 87, 97; and moduli with completely novel prime factors: p=49,127,169). We found that lower balancedness again yielded better generalization (Fig. 6a); and further, the model performed better on moduli with shared prime factors than moduli with completely novel prime factors (Fig. 6b). This demonstrates that the insights of our theory appear to transfer to a Transformer-based architecture as well.
> 4. **Data Split:** We vary the CIFAR-100 split from 98/2 (pretraining on 98 classes and fine-tuning on 2) to 60/40. Consistent with our CIFAR results, a smaller initialization scale ($\kappa < 1$) improves generalization (Fig. 5c)
>
> Together with the pre-existing ResNet PT+FT experiments, we believe that these novel experiments substantially increase the empirical scope of our paper.
>
> > **Lack of large-scale model validation.**
>
> We hope that the experiments we describe above partially address your concern. That said, we agree that future work should test the impact of balancedness in even larger architectures and will update the paper to discuss this point; we expect that the empirical intuitions our experiments provided will be useful prerequisites for this.
>
> > **Can adaptive initialization strategies be designed from the theory?**
>
> Our theory suggests that initialization should reflect the degree of alignment between PT and FT. For FT tasks where PT and FT largely relies on the same features, a balanced initialization should work best, whereas for tasks where FT  relies only on a subset of PT features (paired, perhaps, with a subset of novel features as well) or where PT is fairly noisy, an imbalanced initialization should work better, as it retains the ability to further refine its features. Similarly, if the pretrained representation is weak (e.g. indicated by a low amount of pretraining data or low validation performance), more adaptive settings (negative $\lambda_{PT}$) help the model refine or learn new features; for stronger pretraining (i.e. large pretraining data and high validation performance) task overlap will play a more important role in choosing $\lambda_{PT}$ (as discussed above).
>
> Of course, on practical tasks, we may not necessarily know a priori which of these regimes we operate in, highlighting the potential need for cross-validating the choice of balancedness. However, future work could also design more principled techniques for making these choices in nonlinear neural networks, for example by estimating the degree of feature overlap or the sparsity of the learned representation. Thank you for this question; we are quite excited about the potential practical implications of our insights and will add this discussion to the revised version of the manuscript.

---

> > ### Author Rebuttal · Reviewer_r9Q8 · 2026-04-02
> >
> > Thanks for the authors' response, my main concerns have been resolved. I believe such theoretical contribution is valuable,  but further scaling up of the model architecture (e.g., ViT-L, DINO, CLIP) is still necessary for verification in the future.

---

### Official Review · Reviewer_nGfa · 2026-03-11

**Soundness:** 3
**Presentation:** 4
**Significance:** 4
**Originality:** 4
**Overall Recommendation:** 5
**Confidence:** 3

**Summary:**

The paper develops an analytical, end-to-end theory of how pretraining and fine-tuning interact in diagonal linear networks. It derives the implicit inductive bias of fine-tuning as a function of initialization parameters and the pretrained solution βPT, and then uses replica theory to obtain exact high-dimensional predictions for fine-tuning generalization under a teacher–student model with controllable sparsity and task overlap. The analysis reveals four qualitatively distinct fine-tuning regimes and a fundamental trade-off between sparsity bias and pretraining dependence, with theoretical predictions validated by simulations and suggestive experiments on ResNet trained and fine-tuned on CIFAR-100 dataset.

**Compliance With Llm Reviewing Policy:**

Affirmed.

**Final Justification:**

My concerns are addressed and I have no further questions.

**Key Questions For Authors:**

1. How do BN layers interact with your cPT and κ interventions in ResNet-18? Did you freeze BN or otherwise control for its scale invariance, which can neutralize global weight scaling?
2. Can you justify more fully why resetting the effective function to zero after pretraining preserves the essential inductive bias of typical fine-tuning that starts from the pretrained function? Do results change if you initialize fine-tuning from the actual pretrained function without rebalancing?
3. How do your predictions change when αPT < 1 or when σ0 > 0 in pretraining labels? Is there a tractable extension to model partial recovery of βPT and its effect on PD?
4. Your theory assumes gradient flow. Do SGD/Adam and finite step sizes materially alter the observed regimes in practice?
5.  Did you attempt to control or estimate overlap between pretraining and fine-tuning classes, and does the benefit of κ/cPT/γFT manipulation correlate with estimated overlap as predicted?

**Limitations:**

yes

**Strengths And Weaknesses:**

Strengths:
1. Derives a closed-form implicit bias for fine-tuning in DLNs that depends on both initialization scales and the pretrained representation, extending prior analyses beyond small-initialization limits.
2. Identifies and formalizes a four-regime landscape, including a lazy, pretraining-independent regime accessible only in the PT+FT setting.
3. Provides analytic characterization of the ℓ-order/PD trade-off and shows how λPT, cPT, and γFT move the system along different axes of this trade-off.
4. Well-structured exposition separating implicit bias derivation, replica characterization, regime taxonomy, and empirical validations.
5. Addresses how pretraining shapes affects inductive bias and sample efficiency in fine-tuning.

Weaknessnes:
1. The setting αPT ≥ 1 narrows scope; the role of imperfect pretraining or noisy labels (σ0 > 0) is not deeply explored beyond formal inclusion.
2. The rebalancing step that resets βFT(0) = 0 after pretraining which is nonstandard in practice; while invariants justify the induced penalty, this choice may limit external validity for realistic pipelines that start fine-tuning from the pretrained function.
3. The mapping from theoretical parameters to ResNet interventions specially, λPT ↔ κ scaling of early stages, and cPT scaling in the presence of batch normalization, needs more careful justification as BN can neutralize absolute scaling.
4. Large-scale experiments focus on two-class fine-tuning on CIFAR-100 with random splits; the “overlap” structure is not controlled or measured, making it hard to connect to the spike-and-slab overlap parameters.

---

> ### Author Rebuttal · Authors · 2026-03-31
>
> We thank the reviewer for their thoughtful comments. We link to additional experiments (Figures) discussed in the rebuttal [here](https://shorturl.at/c11PL).
>
> > **How do BN layers interact with your cPT and κ interventions in ResNet-18?**
>
> We did not freeze the Batch Normalization (BN) layers. Instead, we designed our scaling intervention to explicitly account for BN's scale invariance. Our $\kappa$ and $c_{PT}$ intervention scales all learnable parameters within each of the first three blocks, including the BN affine parameters $\gamma$ and $\beta$.
> Because BN's $\gamma$ is scaled by the same factor, the effective output of each block is scaled by  $\kappa$ and $c_{PT}$. As a result, the scaling persists in the block's output activations and is not absorbed by normalization.
>
> > **Justify resetting the effective function to zero after pretraining**
>
> We reset the effective function to zero because we wanted to focus on the task overlap in terms of dimensions. Without this rebalancing step, an additional inductive bias towards features of the same sign as during PT would arise. This is an artifact of the specific way in which diagonal linear networks are structured, as the signs within each pathway are conserved. Rebalancing therefore moves us into a more typical FT regime, and allows us to focus on the impact of overlapping dimensions (in line with Lippl & Lindsey, 2024). We note that for $\gamma_{FT}=0$, the network would exhibit the same inductive bias without rebalancing, though $\beta_{aux}$ would have to be divided by two. More broadly, without rebalancing, the same intuitions on how the different initialization parameters impact the learning regimes will apply (with an additional same-sign inductive bias, as mentioned above). We will add this discussion to the manuscript and appreciate you asking this question, as we agree that this design choice merits further explanation.
>
> > **How do your predictions change when αPT < 1**
>
> To address this point, we extended our analysis to the imperfect-PT regime $\alpha_{PT}<1$ and derived the corresponding generalization error curves using replica theory, allowing us to model partial recovery of $\beta_{PT}$ analytically.
>
> The key effect is that $\alpha_{PT}$ controls how strongly PT imprints structure, while $\lambda_{\mathrm{PT}}$ determines whether this structure is reused or adapted during FT. This is illustrated in Fig.3 and Fig.4 in the attached pdf. In Fig.3 (varying overlap), when $\alpha_{PT}$ is small, behavior is largely insensitive to $\lambda_{PT}$, as the PT signal is weak; as $\alpha_{PT}$ increases, $\lambda_{PT}$-dependent regimes emerge, particularly in the no-overlap setting where adaptation is required. In Fig.4 (varying sparsity), for sparse FT tasks the effect of $\lambda_{PT}$ is visible even at small $\alpha_{PT}$, whereas for denser tasks it only emerges once $\alpha_{PT}$ is sufficiently large. Notably, while for $\alpha_{PT}=1$, negative $\lambda_{PT}$ makes generalization worse for full overlap (as it leads to decreased pretraining dependence compared to $\lambda_{PT}=0$, Fig. 3a in the main paper), for $\alpha_{PT}=0.2$, it actually improves performance; this is because it alleviates the consequences of imperfectly learning the pretraining structure and allows the model to further refine its features during FT (Fig. 3 in the new document). This highlights that imbalanced initialization may be even more important for tasks with smaller PT datasets. Overall, these new results provide novel insight into PT+FT under imperfect pretraining; thank you for your suggestions.
>
> > **Your theory assumes gradient flow. Do SGD/Adam and finite step sizes materially alter the observed regimes in practice?**
>
> Please see our response to Reviewer eu24 for a detailed discussion. Briefly, although our theory is derived under gradient flow, the same qualitative regimes persist across SGD and Adam variants: negative $\lambda_{PT}$ consistently improves performance in adaptation-heavy and sparse settings, and these trends are robust to finite step sizes, stochasticity, and adaptive updates.
>
> > **Did you attempt to control or estimate overlap between pretraining and FT classes, and does the benefit of κ/cPT/γFT manipulation correlate with estimated overlap as predicted?**
>
> Yes—while we do not directly quantify task overlap, we investigate its impact by FT on a different dataset (SVHN), which is less similar to the PT data and therefore serves as a proxy for lower overlap (newly added results; see Fig. 5a). FT performance exceeds that of single-task learning at intermediate training stages across all kappa values. Consistent with our CIFAR results, initializing ResNet with a smaller-than-standard scale ($\kappa < 1$) enhances FT generalization. Overall, these findings support the hypothesis that reducing dependence on PT amplifies the advantages of $\kappa<1$ scaling for tasks with lower overlap (compared to the CIFAR experiment in Fig. 4 of the main paper).

---

> > ### Author Rebuttal · Reviewer_nGfa · 2026-04-04
> >
> > I thank the authors for their detailed and comprehensive rebuttal and hope their contributions impact the advancement of scientific community.

---

### Official Review · Reviewer_eu24 · 2026-03-13

**Soundness:** 3
**Presentation:** 3
**Significance:** 1
**Originality:** 2
**Overall Recommendation:** 4
**Confidence:** 3

**Summary:**

This paper explores on how initialization strategies dictate whether a model simply reuses features or actually learns new ones.

The authors use diagonal linear networks to derive generalization error based on task statistics and initialization scales.

Their core contribution is the identification of four distinct fine-tuning regimes: with either rich or lazy and either pretraining-independent or dependent.

The discovery that the relative scaling of weights between early and late layers is the primary lever controlling these behaviors.
They find that a small initialization in early layers allows for a pretraining-dependent rich regime that maximizes the benefit of transferred knowledge while still allowing the model to adapt.

**Compliance With Llm Reviewing Policy:**

Affirmed.

**Final Justification:**

My concerns are addressed in rebuttal. I recommend this paper to be accepted.

**Key Questions For Authors:**

will the pretraining-dependent rich regime remains stable when moving beyond diagonal linear networks? You don't have to prove anything, but I do want to see some analysis on this point, which could build up the understanding of the readers on those issues.

How sensitive the methods to the optimizer choice?

**Limitations:**

I failed to find discussion on limitations

**Strengths And Weaknesses:**

Soundness: The paper is mostly technically sound.
Presentation: The paper is will structured and mostly clearly. Although the high density of mathematical notation and specialized physics-based proofs is challenging to read.

Significance: The analysis is mostly based on linear models, which may not fully capture the complex hierarchical feature interactions found in non-linear architectures. the significance is limited.

Originality: I believe the the 4 regimes are now. But a lot of intermediate steps are heavily on established diagonal network frameworks

---

> ### Author Rebuttal · Authors · 2026-03-31
>
> We thank the reviewer for their thoughtful comments. We link to additional experiments (Figures) discussed in the rebuttal [here](https://shorturl.at/c11PL).
>
> > **Significance**:
>
> As highlighted in the literature (e.g. Nam et al., 2025, Saxe et al., 2014), linear networks are often dismissed as lacking expressive power. However, despite their apparent simplicity, their training dynamics are inherently non-linear. This non-linearity allows them to capture and explain a range of key phenomena observed in modern deep learning. These phenomena include emergence in large language models (Brown et al., 2020; Nam et al., 2024), neural collapse in image classification (Mixon et al., 2020; Papyan et al., 2020), lazy and rich training regimes (Jacot et al., 2018; Chizat et al., 2019), and grokking (Kunin et al., 2024; Power et al., 2022). Our work extends this research. To demonstrate that our theoretical insights extend to nonlinear networks as well, we show that our results are robust across different nonlinear model architectures (Transformers and ResNet across multiple optimisers, and data splits. Fig. 5 and 6, see our response to reviewer r9Q8).
>
> > **Originality**:
>
> We agree that our paper builds on a prior line of work on diagonal linear networks, but emphasize that the systematic investigation of specific models over multiple papers is crucial to a detailed understanding of phenomena in deep learning. Moreover, a core contribution of our paper is that we provide a full-stack theoretical characterization of the inductive bias of diagonal linear networks: we describe how their weights evolve over PT and FT and further leverage replica-theoretical tools to describe the resulting generalization error across a family of tasks. To our knowledge, prior work has not provided such analysis in a PT+FT setup. This integrated view offers deeper insights than norm inspection alone; for instance, our theory highlights the critical role of balancedness in the PT-dependent rich regime (Fig. 3f of the main text). We appreciate this feedback and will clarify these points in the contributions section.
>
> > **Pretraining-dependent rich regime beyond diagonal networks?**
>
> To provide an intuition for why diagonal linear networks may shed light on nonlinear neural networks, we note that the learning dynamics in ReLU networks can be decomposed into two orthogonal directions: one direction along which the normalized weight vector in the first layer changes, and one direction along which the magnitude of the first and second layer changes. The learning dynamics in the latter subspace are identical to those of a diagonal linear network. Thus, while the changes in the normalized weight vector may induce divergences in the learning dynamics, these different updates have a direct mathematical connection, which grounds their potential relation.
> Beyond simple ReLU networks, we analyzed whether changes in the initialization scale in large-scale ResNets induced the predicted changes, by measuring representation dimensionality before and after training (smaller dimensionality being indicative of richer learning dynamics, Farrell et al., 2023). We found that smaller relative and absolute scale decreased the dimensionality after FT, consistent with our prediction that this should induce richer FT dynamics (Figs. in Appendix D). Your query prompted us to move a detailed analysis and compressed figures connecting our theoretical model to practical nonlinear networks into the main text, and to add a paragraph discussing the mathematical link between diagonal linear and ReLU network dynamics.
>
> > **Method robustness to optimizer choice**
>
> In diagonal linear networks, we compare SGD, Adam with weight decay, and Adam with L2SP regularization (Li et al, 2018). In Fig. 1 of the linked pdf (varying overlap), all optimizers exhibit the same task-dependent behavior: negative $\lambda_{\mathrm{PT}}$ improves performance when adaptation is required (low overlap), while differences shrink when reuse is sufficient (high overlap). In Fig. 2 (varying sparsity), negative $\lambda_{\mathrm{PT}}$ is the only setting that consistently benefits from sparsity across all optimizers, matching the predicted PT-dependent rich regime. We observe the same qualitative trend in nonlinear settings: in Fig. 5b, using a ResNet trained with Adam, smaller $\kappa$ (our practical proxy for negative relative scale) consistently improves FT performance, indicating that the effect is not tied to a specific optimizer or model. While different optimizers introduce stochasticity, finite step sizes, and adaptive scaling, the qualitative dependence on relative scale and task structure remains unchanged, indicating that the observed regimes are not optimizer-specific.
>
> >**Limitations**
>
> Thank you for pointing this out. We agree we should provide a broader discussion and will move our discussion of limitations from the appendix (Section D.5, line 1090) into the main text.

---

> > ### Author Rebuttal · Reviewer_eu24 · 2026-04-04
> >
> > My concerns are addressed. I raise my rating to 4.

---

### Decision · Program_Chairs · 2026-04-30

**Decision:**

Accept (regular)

**Comment:**

# Summary

This paper studies a pretraining–fine-tuning scenario in diagonal linear networks, exploring how initialization strategies determine whether the trained model will reuse features learned during pretraining or learn new features through finetuning. Depending on different initialization choices, the paper presents four different regimes based on the model’s tendency to reuse vs. not reuse pretrained features, and learn vs. not learn new features. The authors find that a small initialization in early layers allows for a rich regime that depends on pretraining and maximizes the benefit of transferred knowledge while still allowing the model to adapt.

# Comments

Although the paper considers simplified models and tasks, it delivers useful insights into feature reuse/refinement in practical PT+FT scenarios and how best to select initialization scales. The authors addressed most concerns raised in the reviews, and the evaluations are unanimously positive. I recommend acceptance of this paper.

I would like to recommend that the authors incorporate the points raised during the discussion in the revised version. In particular, it would be beneficial to include the additional experimental results as well as the $\alpha_{PT} < 1$ analysis. Also, the authors should carefully discuss why setting $\beta_{FT}(0) = 0$ at the beginning of fine-tuning can be justified. As Reviewer PQKQ also noted, I believe that the paper has some room for improvement in terms of clarity. Please make the promised revisions. Providing more background on replica theory and the implications of Proposition 4.2 would also help.